# Relating pathogenic loss-of-function mutations in humans to their evolutionary fitness costs

Ipsita Agarwal[1,2]*[†], Zachary L Fuller[1][†], Simon R Myers[2,3], Molly Przeworski[1,4]

[1]Department of Biological Sciences, Columbia University, New York, United States; [2]Department of Statistics, University of Oxford, Oxford, United Kingdom; [3]The Wellcome Centre for Human Genetics, University of Oxford, Oxford, United Kingdom; [4]Department of Systems Biology, Columbia University, New York, United States

**Abstract** Causal loss-of-function (LOF) variants for Mendelian and severe complex diseases are enriched in 'mutation intolerant' genes. We show how such observations can be interpreted in light of a model of mutation-selection balance and use the model to relate the pathogenic consequences of LOF mutations at present to their evolutionary fitness effects. To this end, we first infer posterior distributions for the fitness costs of LOF mutations in 17,318 autosomal and 679 X-linked genes from exome sequences in 56,855 individuals. Estimated fitness costs for the loss of a gene copy are typically above 1%; they tend to be largest for X-linked genes, whether or not they have a Y homolog, followed by autosomal genes and genes in the pseudoautosomal region. We compare inferred fitness effects for all possible de novo LOF mutations to those of de novo mutations identified in individuals diagnosed with one of six severe, complex diseases or developmental disorders. Probands carry an excess of mutations with estimated fitness effects above 10%; as we show by simulation, when sampled in the population, such highly deleterious mutations are typically only a couple of generations old. Moreover, the proportion of highly deleterious mutations carried by probands reflects the typical age of onset of the disease. The study design also has a discernible influence: a greater proportion of highly deleterious mutations is detected in pedigree than case-control studies, and for autism, in simplex than multiplex families and in female versus male probands. Thus, anchoring observations in human genetics to a population genetic model allows us to learn about the fitness effects of mutations identified by different mapping strategies and for different traits.

**\*For correspondence:**
ia2337@columbia.edu

[†]These authors contributed equally to this work

## Editor's evaluation

This paper directly estimates the fitness cost of loss-of-function mutations in almost every gene in the human genome, providing an interpretable measure of the severity of mutations. The authors then compare datasets of presumably healthy individuals and individuals affected by severe complex disorders or genetic disorders, finding enrichment of de novo loss-of-function mutations in highly constrained genes among probands alongside other illuminating results. This important study will be useful to researchers interested in interpreting and prioritizing disease-causing mutations and in the process of human evolution. Overall, the approach is elegant and the results are of high quality and compelling.

## Introduction

The ability to identify genetic variants that may be pathogenic and prioritize among them is central to diagnosing, understanding, and treating human disease. Of particular significance is the class of variants that cause functional knock-outs or knock-downs in genes (i.e., "loss-of-function" variants) and may substantially impact disease risk in their carriers (*MacArthur et al., 2012*). All else being equal, individuals carrying a loss-of-function (LOF) allele that negatively impacts their ability to survive and reproduce in their environment will leave fewer descendants on average, and consequently that LOF allele will be at lower frequency in the population at present day. Therefore, observing a depletion of LOF variants in a gene relative to putatively neutral variants is indicative of their deleteriousness.

This notion motivated the development of a number of measures of 'mutation intolerance' that effectively rank genes by the deficit of LOF variants in large samples (*Petrovski et al., 2013*), notably widely used measures *pLI* (*Lek et al., 2016*) and *LOEUF* (*Karczewski et al., 2020*). Both measures are based on the number of unique LOF variants observed in a gene and the number expected under a mutation model for the gene. *pLI* relies on the average depletion of observed LOF variants in genes annotated as recessive or severely haploinsufficient in the ClinGen dosage sensitivity gene list and a hand-curated gene set of Mendelian disorders to classify genes as 'neutral,' 'recessive,' or 'haploinsufficient' (*Lek et al., 2016*). Genes with a high probability assignment (≥0.9) to the haploinsufficient class are classified as 'extremely loss-of-function intolerant.' *LOEUF* does not rely on a reference gene set and is instead a score between 0 and 2, where 0 indicates greater mutation intolerance. Specifically, the authors assume a Poisson distribution of LOF mutations in a gene and assign an upper 95% confidence limit on the underlying mean number of such mutations as a factor of the expected number of LOF mutations for this gene (*Karczewski et al., 2020*). Genes classified as highly 'mutation intolerant' by these measures are enriched for variants that lead to Mendelian genetic diseases (e.g., *Beck et al., 2020*; *Chopra et al., 2022*; *Hansen et al., 2019*; *Oved et al., 2020*; *Timberlake et al., 2019*). A number of recent papers report an enrichment of variants in 'mutation-intolerant genes' for severe complex disease risk as well (e.g., *Antaki et al., 2022*; *Cappi et al., 2020*; *Feng et al., 2019*; *Liu et al., 2020*; *Palmer et al., 2022*; *Sanders et al., 2019*; *Satterstrom et al., 2020*; *Singh et al., 2022*; *Wilfert et al., 2021*; *Zoghbi et al., 2021*). In turn, *pLI* and *LOEUF* are often relied on to classify unknown variants in terms of their likely pathogenic effects (e.g., *Gudmundsson et al., 2022*; *Lee et al., 2022*; *Qi et al., 2021*; *Sharo et al., 2022*; *Wang and Li, 2020*).

Measures such as *pLI* and *LOEUF* implicitly assume an underlying population genetic model of mutation-selection balance (*Cassa et al., 2017*; *Fuller et al., 2019*). Viewing them in light of this model clarifies that they reflect fitness effects over evolutionary time scales, rather than haploinsufficiency with regard to any particular phenotype (*Fuller et al., 2019*). More precisely, for an autosomal gene, they are proxies of the fitness reduction in heterozygotes relative to individuals with two intact copies, commonly parameterized as *hs* in population genetic models, where *s* is the fitness cost of losing both copies and *h* indicates the extent of dominance in fitness. Assuming that there is some selection against the loss of one copy, in a random-mating population, homozygotes should be too infrequent to appreciably affect allele dynamics (*Charlesworth and Charlesworth, 2010*), and the depletion of LOF variants in a gene will be reflective of the strength of selection acting on heterozygotes, *hs*. The same general reasoning applies to the X chromosome, but with complications, as at most X-linked genes, males are hemizygous and females undergo random X-inactivation. Given the lack of a second copy in males, the sex-averaged fitness cost of a LOF should be higher than on autosomes all else being equal, and X-linked genes are therefore expected to show a greater depletion of LOF variants (*Charlesworth and Charlesworth, 2010*).

Under a model for mutation and genetic drift, the observed depletion of LOF variants can be used to directly infer the parameter *hs*; in fact, under a constant population size model and some models of population size changes, and assuming all LOF variants within a gene have the same fitness effect, the sum of the frequencies of LOF variants in a gene is close to a sufficient statistic for *hs* (see *Fuller et al., 2019*; *Simons et al., 2014*). A pair of recent studies took this approach to estimate *hs* for autosomal genes from ~30,000 individuals, initially under a deterministic approximation (*Cassa et al., 2017*), which neglects the effects of genetic drift and changes in population size (*Charlesworth and Hill, 2019*; *Weghorn et al., 2019*), and subsequently incorporating a plausible model of demographic history (*Weghorn et al., 2019*). Recasting measures of gene intolerance in terms of an underlying fitness parameter makes their values more interpretable: whereas a *pLI* value of 0.45 vs. 0.9 has no

clear meaning, doubling the selection coefficient does. Moreover, by specifying the underlying model, different sources of uncertainty can be explicitly incorporated.

As these considerations also make clear, however, estimates of *hs* and proxies like 'measures of intolerance' are reflective of fitness effects over many ancestors, i.e., genetic backgrounds and environments, and many generations. Given how drastically the human environment has changed in the recent past, as well as evidence for variable penetrance of disease mutations (*Cooper et al., 2013*; *Kingdom et al., 2022*), it is unclear what relationship to expect to present-day disease risk. We therefore undertook a systematic examination of the correspondence between the evolutionary fitness costs of LOF mutations and their consequences for developmental disorders and early-onset complex diseases. To this end, we estimated the posterior distributions of *hs* for the loss of a gene copy on autosomes using exome sequences from 55,855 individuals. We also extended the model to different compartments of the X chromosome, taking into account sex differences in mutation and selection, to obtain estimates for X-linked genes. We then used these estimates to learn about the fitness effects of de novo LOF mutations identified in patients for six developmental and neuro-psychiatric disorders.

## Results

### Our estimation approach

For each of 18,282 autosomal and X-linked genes, we estimated the posterior distribution of the fitness cost for heterozygous carriers (*hs*) of LOF alleles using a sequential Monte Carlo Approximate Bayesian Computation (ABC-SMC) approach (*Figure 1A*; see *Supplementary file 2* for these estimates, and analogous ones for the X-chromosome). To this end, we simulated a Wright–Fisher population forward in time in order to generate the frequency of LOF at a gene and compare it to the frequency observed in the Non-Finnish European (NFE) sample of 55,855 individuals in gnomAD (*Karczewski et al., 2020*). We assumed that LOF alleles arise at a mutation rate μ per gene per generation (as described in *Samocha et al., 2014*; *Karczewski et al., 2020*) and that any high-confidence LOF mutation in a gene has the same fitness cost (*Agarwal and Przeworski, 2021*). We also assumed a demographic history for the population, based on the *Schiffels and Durbin, 2014* model (*Schiffels and Durbin, 2014*), which we modified slightly to better match neutral polymorphism levels observed in the NFE sample (see 'Materials and methods'). Proposed values of the dominance coefficient, *h*, and the strength of selection in homozygotes, *s*, were sampled from a uniform and log-uniform prior distribution, respectively (see 'Materials and methods'). Although on autosomes only the compound *hs* parameter can be estimated, we sample *h* and *s* instead of *hs* to enable comparisons between autosomes and the X chromosome. The resulting posterior distribution of *hs* for a gene thus represents the probability of *hs* given the observed LOF frequency, a mutation rate, and a realistic demographic history.

We verified that our choices of mutation and demographic models provide a good fit to observed de novo mutation rates and patterns of neutral polymorphism (*Appendix 1—figures 1–3*, 'Materials and methods'), and that our inference approach allowed us to get robust estimates of simulated posterior distributions (*Appendix 1—figure 4*).

Of the genes considered, a subset (285; *Supplementary file 1*) have observed LOF frequencies that are unusually high under a neutral model (and *a fortiori*, a model with hs > 0). These likely represent cases where our model is misspecified, perhaps because the mutation rate to LOF alleles is in fact higher or due to other biological features (e.g., there is balancing selection on mutations in the gene; see *Amorim et al., 2017*; *Lenz et al., 2016*; *Monroe et al., 2021*). Another possibility is that some mutations are incorrectly annotated as LOF (*Cummings et al., 2020*; *Karczewski et al., 2020*; *MacArthur et al., 2012*). Given these concerns, we excluded these 285 genes from further consideration. Among the remaining 17,318 autosomal genes (*Figure 1B*), the mean maximum a posteriori (MAP) estimate of *hs* is 0.058 while the median is 0.018; in other words, the loss of a gene copy typically inflicts a decrease in fitness of greater than 1%. The data thus provide evidence of strong constraint for many genes: by contrast, the median constraint under the prior is only 0.04%.

Inferred MAP values of *hs* span several orders of magnitude, however, ranging from ~$10^{-6}$ (*GOLGA8S*) to 0.55 (*RIF1*). Overall, there is good agreement between the relative ranks of genes (using our point estimates of *hs*) and a previous estimate of selection coefficients on LOF alleles (*Weghorn et al., 2019*) (Spearman's rank correlation = 0.82) (*Appendix 1—figure 5*). The point

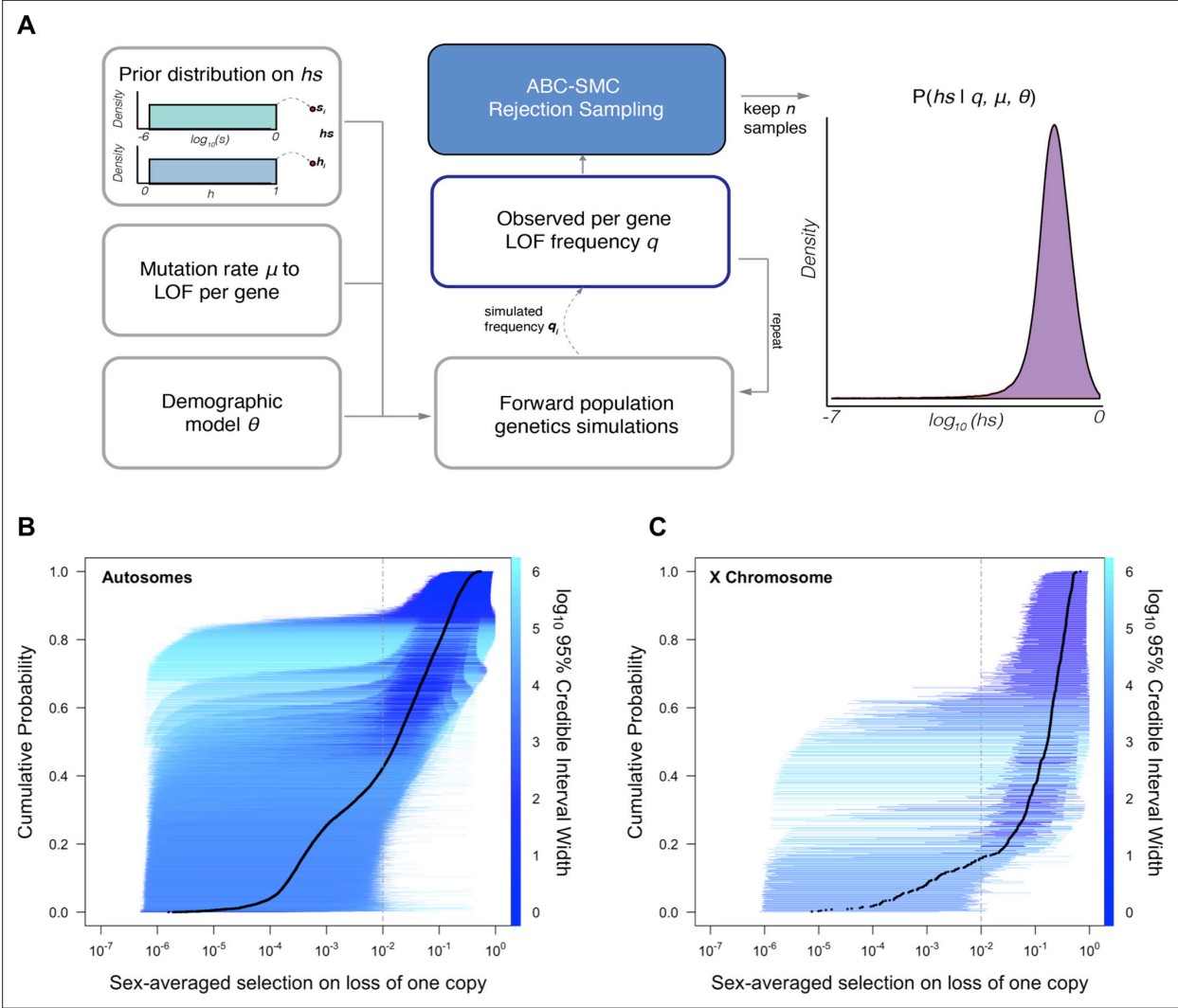

**Figure 1.** Estimating *hs* for loss-of-function (LOF) across human genes. (**A**) Schematic of the approach to infer heterozygous selection coefficients (*hs*) for each gene. We assume prior distributions $\log_{10}(s) \sim U(-6,0)$ and $h \sim U(0,1)$. We further assume a mutation rate µ to LOF alleles per gene and a demographic model specified by parameters $\theta$, which describe changes in the effective population size $N_e$ at time points in the past. These parameters are used in forward population genetic simulations based on a Wright–Fisher model of selection (see 'Materials and methods'). For each iteration *i*, the simulation generates a frequency $q_i$ of LOF alleles, which is then compared to the observed LOF frequency *q* for a given gene. The proposed value of $(hs)_i$ is retained if within a tolerance ε, which is decreased over time, or rejected otherwise. For each ε, this procedure is repeated until there are 50,000 acceptances, providing a sample from the posterior distribution of the probability of *hs* given the observed frequency *q* of LOF variants for a gene (as well as the mutation rate and demographic model). (**B**) The cumulative distribution of the estimated heterozygous selection coefficient *hs* for each autosomal gene. Black dots represent the point estimate of *hs* for each gene, based on the maximum a posteriori estimate (i.e., the mode) of the posterior distribution. Horizontal lines represent the 95% credible intervals for each gene and are colored according to the width of the interval on a $\log_{10}$ scale. (**C**) A similar plot, but for non-pseudoautosomal region (PAR) X-linked genes, with sex-averaged selection on the loss of a copy on the X calculated as the average of *s* and *hs* (see 'Materials and methods').

estimates themselves are somewhat less congruent (*R* = 0.72); this is to be expected as the previous approach relied on a smaller sample and the grid of selection coefficients led to a ridge of estimates near hs = 0.4% (see **Appendix 1—figure 5**).

As is clear from the posterior distributions, the 95% credible interval of *hs* often spans multiple orders of magnitude. In other words, there is substantial uncertainty around our estimates for any given gene, arising from sampling noise as well as the effects of genetic drift (**Figure 1B**). Even for genes with large point estimates, there can be substantial probability mass on much weaker selection (e.g., $hs < 10^{-4}$): for example, of the 9987 genes for which the point estimate is indicative of strong selection ($hs > 10^{-2}$), ~35% have at least 5% of their probability mass on quite weak selection ($hs <$

10⁻⁴). As a result, whereas based on point estimates alone, it appears that over two-thirds of all autosomal genes in humans are under strong constraint (*Figure 1B*; *Weghorn et al., 2019*), based on summing posterior probabilities of hs > 1% for each gene, only half (48%) are estimated to be highly constrained. Nonetheless, this number is still much higher than the prior likelihood of a gene being highly constrained (of 26%).

## Extension to the X chromosome

The X chromosome plays an important role in a number of human developmental disorders (*Lubs et al., 2012*; *Martin et al., 2021*). Because the number of copies differs between the sexes (outside the pseudoautosomal regions (PARs)), the standard autosomal models for mutation, selection, and drift are not directly applicable to all genes on the X chromosome. On autosomes, all heterozygotes can be modeled as having a fitness cost, *hs,* for the loss of a single gene copy. In contrast, for genes on the X without a functional homolog on the Y chromosome, the mode of selection is sex-specific since LOF of one copy generates a full knockout in males: the fitness cost of the loss of a single gene copy is thus *hs* in females and *s* in males.

We extended our approach to these genes by adjusting our Wright–Fisher simulation framework to account for differences in the mode of selection, as well as differences in inheritance patterns and germline mutation rates between sexes (*Gao et al., 2019*; *Halldorsson et al., 2019*; *Jónsson et al., 2017*). We assumed that a homozygous LOF mutation in females has the same fitness effect as a hemizygous LOF mutation in males (see 'Materials and methods'). In addition to performing the same checks as described above for autosomes, we verified the model for the X analytically under a constant population size (*Appendix 1—figures 1, 2, 3, 6 and 7*; *Charlesworth and Charlesworth, 2010*). Sampling from the same prior distributions on *h* and *s* as described above for autosomes, we estimated the sex-averaged strength of selection (*hs* + *s*)/2, i.e., the average fitness effect of losing one copy in a male or a female, for 660 genes on the X chromosome outside the PAR.

All else being equal, we might expect the sex-averaged strength of selection to be greater for X chromosome genes with no Y homologs compared to autosomes because of stronger selection on the loss of a copy in hemizygous males (*Charlesworth and Charlesworth, 2010*). Such X-linked genes might be under stronger selection in females as well, because of dosage compensation (*Carrel and Willard, 2005*; *Heard and Disteche, 2006*; *San Roman et al., 2021*; *Tukiainen et al., 2017*; *Wainer Katsir and Linial, 2019*). Consistent with this idea, 73% of genes on the non-PAR X are estimated to be under strong selection (i.e., the sex-averaged selection on the loss of one copy is above 1%), whereas only 48% are for autosomes (*Figure 1B and C*; see also *Appendix 1—figure 8A* for a comparison based on point estimates, with p<10⁻¹⁵ by means of a Mann–Whitney *U*-test). These X-linked genes also show more constraint on average than the 19 genes in the PARs, which have two expressed copies in both males and females: of the 19 PAR genes, we estimate that only 14% are under strong selection (see also *Appendix 1—figure 8A*; p=9.9 × 10⁻⁹ for a comparison based on point estimates of *hs* for genes within and outside the PAR on the X).

Less expected are our findings for 16 non-PAR X genes with a Y-chromosome homolog (*San Roman et al., 2021*; see 'Materials and methods'): 93% are estimated to be under strong selection. The loss of one copy of these genes appears to be even more deleterious on average than the rest of the non-PAR X (see also *Appendix 1—figure 8A*; p=9.2 × 10⁻⁴). Thus, the fitness cost of the loss of a gene is higher on X than autosomes whether or not the X-linked gene has a Y chromosome homolog and biallelic expression. As noted by *San Roman et al., 2021*, and suggested by others (e.g., *Park et al., 2010*; *Slavney et al., 2016*), one interpretation may be that rather than sex-biased expression and X-inactivation being the source of greater selective constraint on X-linked genes, differences in gene dosage may be the *consequence* of selection for a sex-specific function.

## The distribution of fitness effects for LOF mutations

Under our assumption that LOF mutations within the same gene have the same *hs,* we can obtain the distribution of fitness effects (DFE) for all possible de novo LOF mutations in the genome by weighting the posterior for each gene by its mutational opportunities to an LOF (see 'Materials and methods'). The area under the DFE indicates that more than 56% of all possible autosomal LOF mutations have an estimated *hs* > 1%, while 20% have an *hs* of 10% or greater (*Figure 2A* shows the result for all autosomal LOF mutations, and *Appendix 1—figure 8B* for the X chromosome).

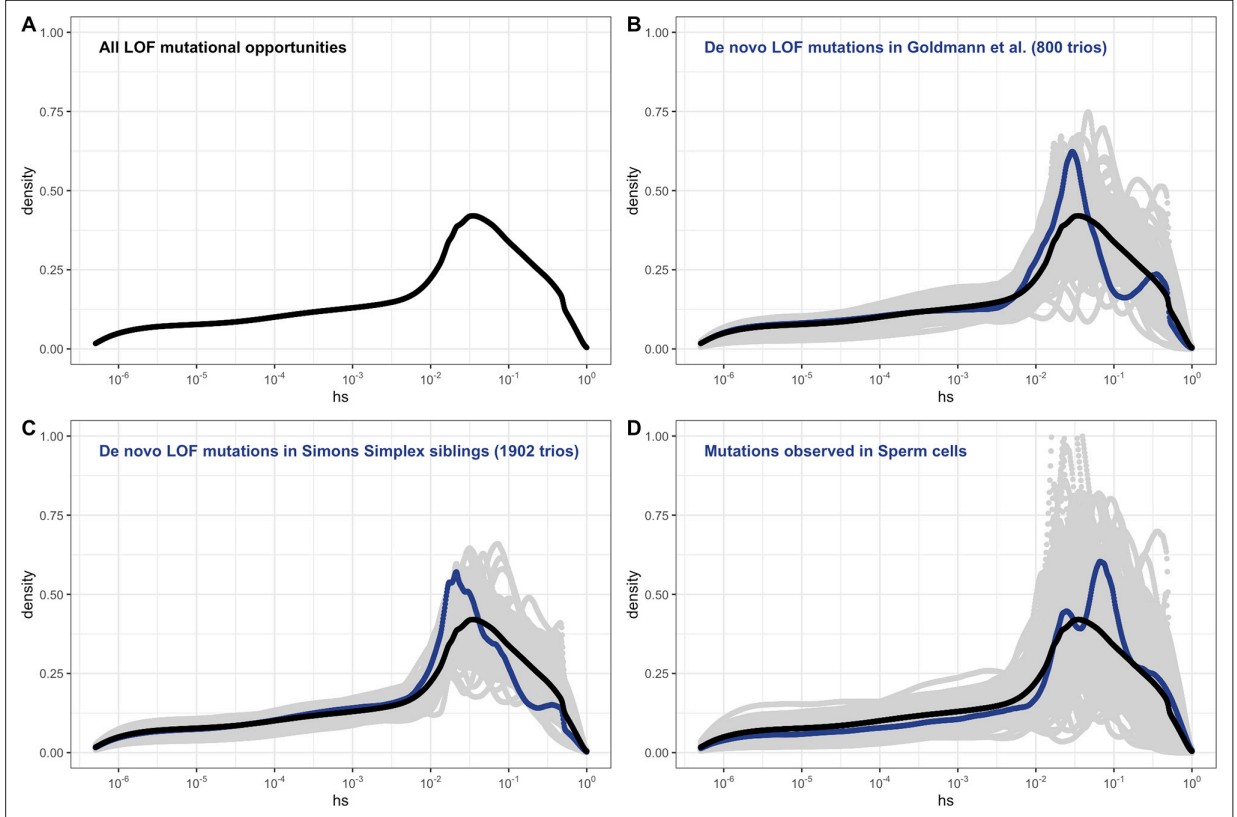

**Figure 2.** The estimated distribution of fitness effects (DFE) across human loss-of-function (LOF) de novo mutations (DNMs) on autosomes, obtained by weighting the posterior distribution of *hs* for each gene with the fraction of potential or observed LOF variants in the gene (see 'Materials and methods'). (**A**) The estimated DFE of all possible de novo LOF mutations in autosomes. The weight assigned to each gene is the fraction of total genome-wide LOF mutational opportunities it contains. (**B**) The estimated DFE of observed de novo LOF mutations (blue curve) in *Goldmann et al., 2016*, obtained by weighting the posterior distribution of *hs* for each gene with the fraction of observed LOF variants it contains, compared to the DFE of all possible LOF mutations (black curve), and 100 bootstrapped DFEs of a set of 37 DNMs randomly sampled from the full set of LOF mutational opportunities (in gray). (**C**) The estimated DFE of observed de novo LOF mutations (blue curve) in the Simons Simplex controls (i.e., unaffected siblings of autism probands), compared to the DFE of all possible LOF mutations (black curve), and 100 bootstrapped DFEs of a set of 64 DNMs randomly sampled from the full set of LOF mutational opportunities (in gray). (**D**) The estimated DFE of new LOF mutations seen in spermatogonial stem cells (blue curve), compared to the DFE of all possible LOF mutations (black curve), and 100 bootstrapped DFEs of a set of 14 DNMs randomly sampled from the full set of LOF mutational opportunities (in gray).

De novo mutations (DNMs) to LOF are sampled from the set of all possible mutations to an LOF. Therefore, the DFE of de novo LOF mutations identified in a representative sample of human pedigrees should approximate the inferred DFE of all mutational opportunities, other than those at which mutations lead to embryonic lethality. With this in mind, we examined the DFE of de novo LOF mutations in a hospital cohort of newborns not ascertained for any disease (*Goldmann et al., 2016*) as well as in unaffected siblings in the Simon Simplex autism study (*An et al., 2018*). Since neither study reported DNMs on the X, we focused on the autosomal DFE, weighting the posterior for each gene by the fraction of observed de novo LOF mutations in that gene. In both cohorts, the DFE of DNMs does not differ significantly from the DFE of all possible LOF mutations (*Figure 2B and C*). The same is observed for the set of LOF mutations seen in spermatogonial stem cells (*Moore et al., 2021*; *Figure 2D*); as these mutations are not ascertained on viability of embryos, they should even more faithfully reflect the set of all possible DNMs.

Although the numbers of mutations are limited, these results suggest that we can treat our estimated DFE as reflective of all possible LOF mutations. Moreover, these findings suggest that the contribution of autosomal LOF mutations that are lethal in the embryo or in early development is likely relatively small.

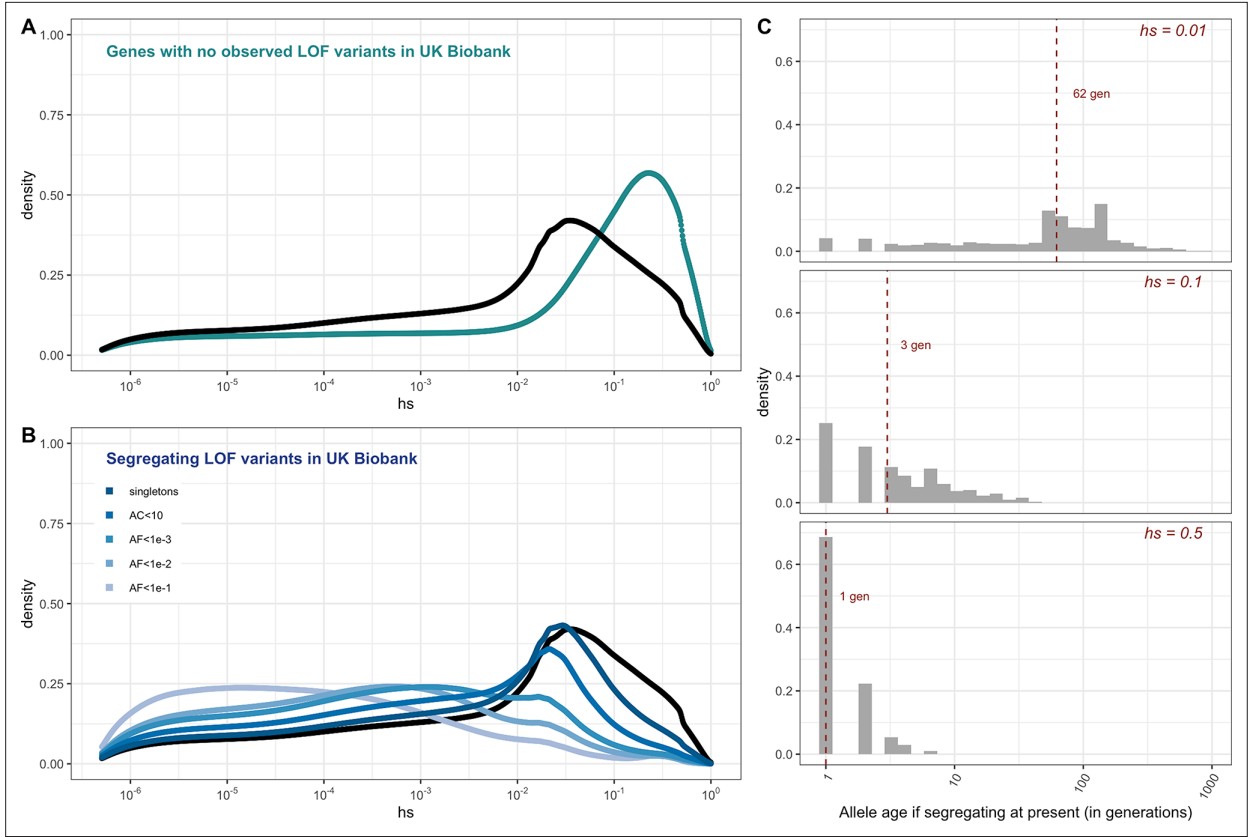

**Figure 3.** The estimated distribution of fitness effects (DFE) for autosomal loss-of-function (LOF) variants in ~160K UK Biobank individuals with sequenced exomes (see 'Materials and methods'). (**A**) The DFE of all possible mutational opportunities in genes that do not have a single loss-of-function (LOF) mutation in ~160K UK Biobank individuals. (**B**) The DFE of segregating LOF variants in ~160K UK Biobank individuals, by allele frequency threshold, compared to the DFE of all possible LOF mutations (black curve). (**C**) The distribution of the age (in generations) of a strongly selected LOF allele segregating in the population at present, obtained using forward simulations at an autosomal locus under a demographic model for population growth in Europe (see 'Materials and methods'), for a heterozygous selection coefficient of hs = 1%; hs = 10% or hs = 50%. The median value in each case is indicated with a red dashed line.

Available data sets on germline mutations identified in human pedigrees indicate that approximately 1 in 1000 de novo mutations in humans lead to an LOF (this estimate does not include the contribution of embryonic lethal mutations) (***Goldmann et al., 2016***). With an average of ~70 DNMs per individual (***Jónsson et al., 2017***; ***Kong et al., 2012***), 1 in ~14 people is therefore born with a DNM that leads to an LOF. Our estimates indicate that at least 20% of LOF are associated with hs > 10%, so roughly 1 in 71 zygotes carry a highly deleterious de novo loss of a gene through a point mutation.

The vast majority of mutations carried by an individual are not DNMs but rather mutations inherited from parents and earlier ancestors. To examine the DFE of segregating LOF mutations, we considered variation data from a population cohort that does not overlap with gnomAD: a subset of 166K individuals from the UK Biobank (***Bycroft et al., 2018***; ***Szustakowski et al., 2020***) . The UK Biobank is a cohort of relatively healthy individuals, who elected to participate at 40–60 years of age (including a small number of individuals with documented diagnoses of schizophrenia and intellectual disability; ***Kingdom et al., 2022***). We focused on a subset of study subjects who are genetically similar to one another and self-describe as White and British (termed 'White British' by the UK Biobank; ***Bycroft et al., 2018***; ***Szustakowski et al., 2020***). Given our coverage criteria and after other filters, we estimate that 6.5% of the point mutation LOFs carried by an individual have an estimated fitness cost of hs >10% (see also ***Appendix 1—figure 9A***). Thus, at least 1 in ~15 humans carries a highly deleterious loss of a gene transmitted by a parent. That individuals who are not diagnosed with severe diseases can nonetheless carry highly deleterious de novo and segregating variants indicates either that even such large effect mutations have variable penetrance, or that carriers have a subclinical but substantial reduction in fertility.

For the set of genes with no LOF variants observed in the UK Biobank sample, mutational opportunities are associated with larger estimated *hs* values than DNMs (*Figure 3A*). In contrast, the DFE of segregating variants is shifted towards lower values of *hs* on average compared to the DFE of possible DNMs. The mean shift in the DFE depends on the allele frequency of segregating variants. In particular, the mean *hs* is higher at lower allele frequencies: singletons in a sample of ~330K (i.e., at frequency one in 330K) chromosomes approach the DFE of observed DNMs and all possible DNMs (*Figure 3B*). These observations follow from first principles since more weakly selected mutations are removed from the population more slowly on average and are more likely to be seen segregating, at higher frequencies on average, than those under strong selection. Accordingly, simulations suggest that if hs = 1%, a mutation sampled in the population at present has persisted for a median of ~60 generations, and if hs = 10%, for a median of only three generations (*Figure 3C*, *Appendix 1—figure 10*, 'Materials and methods').

## The realized fitness burden of LOF alleles underlying severe disease phenotypes

One approach to mapping mutations with a large effect on disease risk is to resequence families with offspring ascertained on the basis of a disease and unaffected parents, and identify DNMs. For severe diseases, LOF mutations are often disproportionately represented among the exonic DNMs identified (e.g., *Deciphering Developmental Disorders Study, 2017*; *Jin et al., 2017*; *Kaplanis et al., 2020*; *Krumm et al., 2015*; *Satterstrom et al., 2020*). A priori, it is unclear what the fitness costs of such LOF mutations should be: notably, they may vary in their penetrance, depending on genetic background and environmental exposures.

We focused on relatively well-defined, severe diseases that manifest early in childhood and are likely to correspond to a substantial realized fitness cost. Specifically, we considered exome data from trios with unaffected parents and probands with one of six clinical diagnoses: developmental disorders; congenital heart disease (CHD); developmental and epileptic encephalopathies; autism; schizophrenia; and Tourette's syndrome or obsessive-compulsive disorder (OCD) (*Cappi et al., 2020*; *EuroEPINOMICS-RES Consortium et al., 2014*; *Fromer et al., 2014*; *Hamdan et al., 2017*; *Howrigan et al., 2020*; *Jin et al., 2017*; *Kaplanis et al., 2020*; *Rees et al., 2020*; *Satterstrom et al., 2020*; *Willsey et al., 2017*; *Xu et al., 2012*). We obtained DFEs for the set of mutations in each disease cohort, as described above (see 'Materials and methods').

If we assume that parents in the pedigree studies have the same genetic ancestries (i.e., similar genomic backgrounds) and experience the same environmental effects as the gnomAD samples used to estimate the DFE of all mutations, then any differences between the DFE of DNMs in probands relative to the DFE for all LOF mutations can be attributed to ascertainment for the disease. In other words, under these assumptions, any shift in the DFE of DNMs in probands reflects a causal contribution of DNMs to the disease diagnosis. In practice, it is very likely that the genetic ancestries of the disease cohorts differ at least somewhat from that of gnomAD; nonetheless, in most cases, inferences of large selection effects should be robust to differences in demographic histories (*Simons et al., 2014*; *Weghorn et al., 2019*).

In the pedigree studies, there is a clear enrichment for mutations with large values of *hs* in cases compared to what is expected for a random sample of de novo LOF mutations in the population (*Figure 4A–F*). For instance, 50% of LOF mutations in the Deciphering Developmental Disorders (DDD) cohort, which consists of individuals with severe developmental disorders, have hs >10%; in comparison, the area under the DFE for a random sample of LOF mutations is only about 20%. A significant enrichment of highly deleterious mutations is observed for the four other diseases examined, all but Tourette's syndrome and OCD. On the X chromosome, there is a similar enrichment of LOF mutations with hs > 10% in the study of developmental disorders (*Appendix 1—figure 8C*); for other diseases, we do not have sufficient data for the X. Thus, the mutations that distinguish individuals ascertained for severe disease from a more representative sample are highly deleterious. At the same time, such mutations do not appear to be fully penetrant in that they are also carried by individuals in the UK Biobank who self-report as healthy (*Appendix 1—figure 9B*; 'Materials and methods').

The degree to which cases are enriched for highly deleterious mutations varies by disease, as can be seen by contrasting the findings for developmental disorders with those for schizophrenia ($p \ll 10^{-5}$, 'Materials and methods'), or with Tourette's syndrome and OCD ($p \ll 10^{-5}$, 'Materials and methods'),

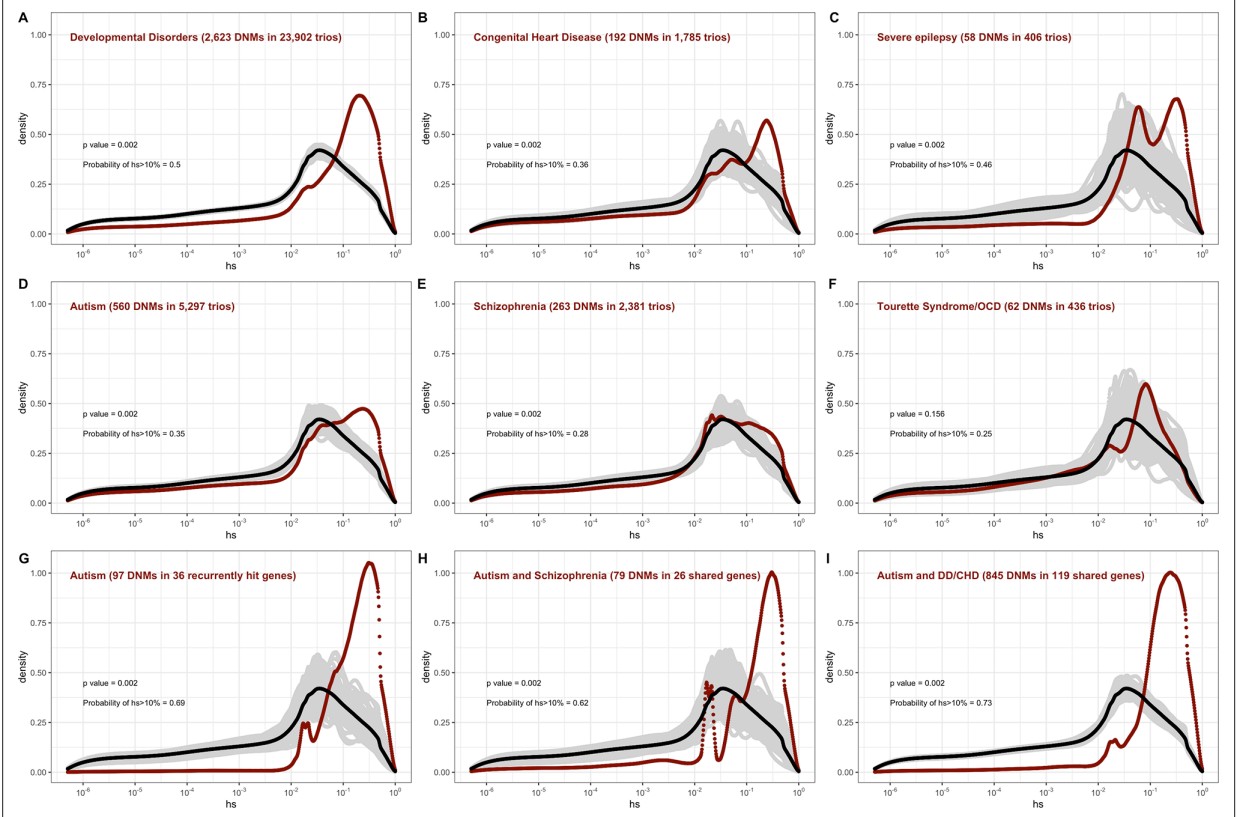

**Figure 4.** The estimated distribution of fitness effects (DFE) for de novo loss-of-function (LOF) mutations seen in individuals affected by severe diseases, from whole-exome-sequenced parent–offspring trios. The DFE for each disease cohort is obtained by weighting the posterior density of *hs* for each autosomal gene with the fraction of observed LOF mutations in the gene in that cohort. In each panel, the DFE of all possible LOF mutations is denoted with a black curve. For *n* de novo mutations (DNMs) in a disease cohort, the gray lines denote 100 bootstrapped DFEs of a set of *n* DNMs randomly sampled from the full set of LOF mutational opportunities. P-values were calculated from the rank of the mean of the distribution for each disease compared to the means of 1000 bootstrapped distributions (see 'Materials and methods'). The probability of hs >10% for all possible autosomal LOF mutations is 20%. The probability of hs > 10% in each panel denotes the area under the distribution (in red) in the interval (0.1,1) for de novo LOF mutations seen in the corresponding disease cohort. The estimated DFE of observed de novo LOF mutations in individuals affected by (**A**) developmental disorders, from the Deciphering Developmental Disorders (DDD) cohort (*Kaplanis et al., 2020*), (**B**) congenital heart disease (*Jin et al., 2017*), (**C**) developmental and epileptic encephalopathy (*EuroEPINOMICS-RES Consortium et al., 2014*; *Hamdan et al., 2017*), (**D**) autism, from the Autism Sequencing Consortium (ASC) and Simons Simplex (SSC) (*Satterstrom et al., 2020*), (**E**) schizophrenia (*Fromer et al., 2014*; *Howrigan et al., 2020*; *Rees et al., 2020*; *Xu et al., 2012*), and (**F**) Tourette's syndrome and/or obsessive-compulsive disorder (OCD) (*Cappi et al., 2020*; *Willsey et al., 2017*). The estimated DFE of observed de novo LOF mutations in genes (**G**) recurrently hit in individuals with autism, (**H**) shared between individuals with autism and schizophrenia, and (**I**) shared between autism and developmental disorders (DD) or congenital heart disease (CHD).

for example. These differences in the DFEs across diseases likely reflect, at least in part, the genetic architecture of the disease (e.g., how many causal mutations of large effect there are), and, relatedly, how correlated the disease phenotype is to fitness. Roughly ordering the diseases by their typical age of onset as a proxy of severity, we see that for more severe diseases, a higher fraction of DNMs are LOF and the LOF mutations identified are more deleterious (*Figure 4A–F*, *Appendix 1—table 1*).

The DFE of DNMs identified in offspring ascertained for disease is a mixture of the DFE for mutations that are causal and mutations that do not contribute to risk. The 2.5-fold enrichment of LOF mutations with hs > 10% due to ascertainment on developmental disorders implies that a DNM identified in a gene with an estimated hs > 10% has a ~60% ( = (2.5–1)/2.5) chance of being causal. More generally, given a set of DNMs mapped in a severe disease cohort, evolutionary fitness cost can be used to prioritize mutations most likely to contribute to disease risk. Again roughly ordering the diseases by their average age of onset, highly deleterious mutations are more likely to be causal for diseases that are expected to arise in development or early childhood than for those with a typical onset in adolescence or early adulthood (*Figure 4A–F*, *Appendix 1—table 1*).

Genes reported as having mutations in multiple probands or studies are more likely to harbor causal mutations. Accordingly, if we consider only genes that have more than one LOF mutation in any of the autism cohorts (*Figure 4G*), there is an almost twofold enrichment of hs > 10% mutations compared to all LOFs seen in autism (*Figure 4D*). This observation suggests that, as expected, when more than one de novo LOF mutation has been found in the same gene in small numbers of pedigrees ascertained for a disease, those LOF mutations are more likely to be causal. Interestingly, a similarly high enrichment of highly deleterious mutations is seen when conditioning on genes that overlap between autism and schizophrenia cohorts, and autism and developmental disorders (*Figure 4H and I*). The explanation may be similar: a gene with two or more independent LOF events in pedigrees ascertained for two different diseases may be more likely to be causal for at least one. But it may also be that an LOF mutation that increases the risk of multiple types of disease or leads to a more severe disease state encompassing multiple syndromes tends to be more severe in its fitness effects.

We further considered case-control studies of autism, schizophrenia, developmental epilepsy, and bipolar disorder (*Feng et al., 2019*; *Palmer et al., 2022*; *Satterstrom et al., 2020*; *Singh et al., 2022*) to examine the DFEs of rare variants in cases and controls (where rarity is defined by the original study; *Appendix 1—figure 11*). Among such variants, cases show only a small enrichment of highly deleterious variants over controls, which is statistically significant for autism, epilepsy, and schizophrenia. These findings are expected: given that a non-negligible fraction of controls harbor highly deleterious alleles (~6.5% in a relatively healthy cohort; *Appendix 1—figure 9A and 11*), a large fraction of cases would have to carry such mutations for the enrichment to be appreciable. Moreover, almost all of the mutations compared between cases and controls are inherited rather than de novo, so have lower *hs* on average (see *Figure 3*). These findings underscore that for a given disease, the DFE of the mutations discovered depends on the design of the mapping study.

## The impact of study design on the DFE of disease mutations

The fitness effects of mutations that underlie a disease phenotype may differ depending on the sex of the proband and the parental background. We examined whether the DFEs of mapped mutations reveal such differences, focusing first on developmental disorders (DD), which have well-defined diagnostic criteria, and where most cases are sporadic rather than familial (*Deciphering Developmental Disorders Study, 2017*; *Kaplanis et al., 2020*). We considered the 7500 trios in the DDD study for which we had information about the sex of the proband (see 'Materials and methods'). The DFE for de novo LOF mutations in affected males is very similar to the DFE for mutations seen in affected females (*Figure 5A and B*), and to the DFE for the full sample of 24K trios with developmental disorders (*Figure 4A*).

In contrast, for autism, the DFE varies markedly by cohort and by sex (*Figures 4D and 5C–J*). To tease apart the effects of different ascertainment criteria, we consider three nonoverlapping cohorts of individuals ascertained for autism and for which information on the sex of the probands is available, namely, the Simons Simplex (*An et al., 2018*; *Fischbach and Lord, 2010*), SPARK (*Feliciano et al., 2019*), and MSSNG (*C Yuen et al., 2017*). Notably, each of these three cohorts has a different proportion of families that are simplex versus multiplex, ranging from almost no families expected to be multiplex in the Simons Simplex cohort, 10% in SPARK, and almost 40% of families in MSSNG (*An et al., 2018*; *C Yuen et al., 2017*; *Feliciano et al., 2019*; *Fischbach and Lord, 2010*). Comparing these cohorts allows us to examine the influence of the parental background, the sex of the offspring, and the two together on the DFE. Consistent with the notion that large effect mutations underlie sporadic cases and a shared oligogenic or polygenic background contributes more to risk in familial cases (e.g., *Antaki et al., 2022*; *Wilfert et al., 2021*), there is a shift of the DFE to smaller *hs* values among DNMs mapped in multiplex versus simplex families (*Figure 5I and J*). In other words, de novo LOF mutations in simplex cohorts are on average more deleterious than those in cohorts that contain multiplex families. These patterns could also reflect differences in disease severity or phenotype definition between family designs.

Further, although the vast majority of probands are male, affected female individuals in Simplex cohorts carry mutations that are much more deleterious (p<<10⁻⁵, *Appendix 1--figure 12*; *Figure 5C–H*). Indeed, a de novo LOF mutation with hs > 10% seen in female cases of simplex autism is on average 1.2–1.5 times more likely to be causal than a similar mutation in males (*Appendix 1—table 1*). This finding is consistent with a 'female protective effect' in autism (*Jacquemont et al., 2014*; *Robinson*

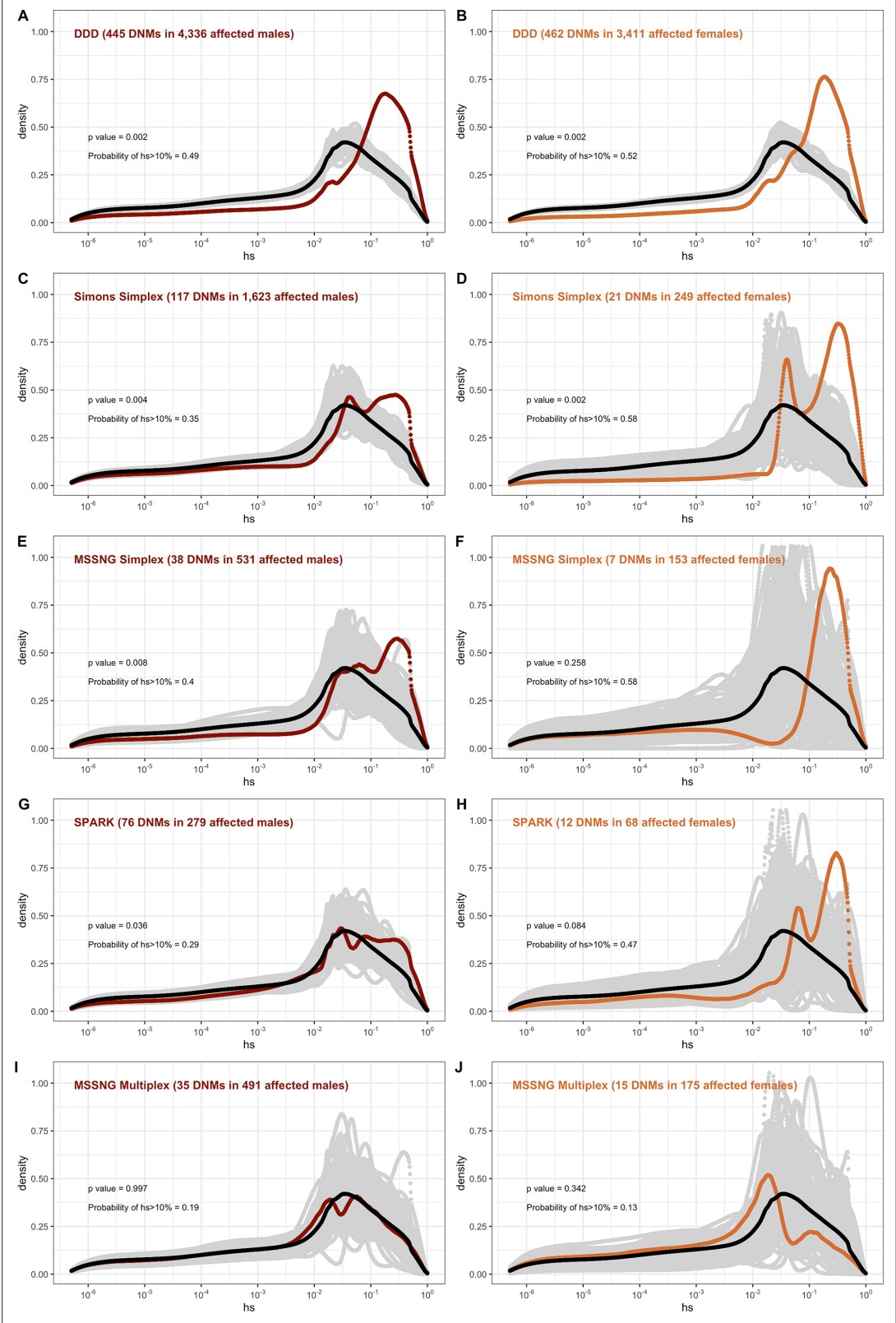

**Figure 5.** The effect of study design and composition on the fitness effects of de novo mutations (DNMs) seen in developmental disorders and autism. In each panel, the distribution of fitness effects (DFE) of all possible loss-of-function (LOF) mutations is denoted with a black curve. For *n* DNMs in a disease cohort, the gray lines denote 100 bootstrapped DFEs of a set of *n* DNMs randomly sampled from the full set of LOF mutational opportunities. The estimated DFE of de novo LOF mutations in (**A**) affected males from the Deciphering Developmental Disorders (DDD) cohort, (**B**) affected females

*Figure 5 continued on next page*

*Figure 5 continued*
from the DDD cohort, (**C**) affected males from the Simons Simplex cohort, (**D**) affected females from the Simons Simplex cohort, (**E**) affected males from the MSSNG cohort, excluding multiplex families, (**F**) affected females from the MSSNG cohort, excluding multiplex families, (**G**) affected males from the SPARK cohort, (**H**) affected females from the SPARK cohort, (**I**) affected males in multiplex families from the MSSNG cohort, and (**J**) affected females in multiplex families from the MSSNG cohort.

*et al., 2013*; *Satterstrom et al., 2020*; *Wigdor et al., 2022*), for instance, if compensation through socialization leads to sporadic autism diagnoses only in females with very severe disease; alternatively, it may reflect a physiological difference in how the disease develops in the two sexes, e.g., through differential effects of sex hormones in development (*Ferri et al., 2018*; *Werling, 2016*). Intriguingly, a sex difference in DNMs is not detected in multiplex families (*Figure 5I-J*), potentially because the disease risk tends to be polygenic in both sexes in such families; in principle, it could also result from females being diagnosed at lower severity thresholds as affected siblings of male probands.

For schizophrenia studies, in turn, there is almost no discernible difference between males and females (*Appendix 1—figure 13*). While we lack information on simplex and multiplex families, cases that have a documented family history of mental illness show a shift towards less deleterious DNMs compared to cases without one (*Appendix 1—figure 13*). As with studies of autism, these findings highlight that for a given disease, commonly varying characteristics of individual cohorts influence the severity of variants discovered. Accordingly, these characteristics impact their utility in elucidating the pathophysiology of the disease.

## Discussion

We estimated the fitness cost of the loss of a single copy for 17,318 autosomal genes, and for the first time, for 679 X-linked genes, based on a model with sex differences in mutation and selection. Posterior modes are presented in *Supplementary file 2*, along with 95% credible intervals, allowing the support for strong selection on any given gene to be assessed, and uncertain estimates to be revisited in light of accumulating data.

As our approach relies on a full generative model of the evolutionary process over hundreds of thousands of generations, we make explicit choices about demography and mutation rates. When the impact of drift is not negligible or there are strong departures from random mating, estimates of selection are sensitive to the choice of the demographic model (*Simons and Sella, 2016*; *Weghorn et al., 2019*). Similarly, systematic error in LOF mutation rates due to a misspecified mutation model or mis-annotation of LOF sites in the genome would bias estimates of selection (linearly for strongly selected genes; *Simons et al., 2014*), and random error in mutation rates would increase the uncertainty associated with estimates of selection. We use a standard model for mutation rate (*Karczewski et al., 2020*) and check that it fits DNM data reasonably well in aggregate (*Appendix 1—figure 1*); we also check that our demographic model fits the data for synonymous variants (*Appendix 1—figure 2*). Nonetheless, explicitly modeling uncertainty in both parameters would be a useful extension of this work.

Our approach also requires us to specify a prior on the fitness effects of an LOF mutation. We chose a prior that is close to uninformative on the order of magnitude of the strength of selection. For genes with little information, that may mean credible intervals that span several orders of magnitude; for some genes, which are known independently to be functionally important, this may be unrealistic. One possibility might be to use an empirical Bayes approach, first pooling information across all genes to obtain a prior, and then inferring posteriors for individual genes. Another natural extension would be to use functional information such as expression levels, and number of protein interactions to specify gene-specific priors, or priors for categories of genes (some examples of this approach already exist for plants, e.g., *Ramstein and Buckler, 2022*). Beyond short genes, we expect such extensions to have most impact on the interpretation of genes under very strong selection: in this approach, we rarely sample *hs* values very close to 1, potentially underestimating the number of dominant lethal genes. With a more informative prior, this set may be more reliably identified.

We make a number of other simplifying assumptions: for instance, we treat compound heterozygotes as homozygotes, which may not be valid (*Clark, 1998*), and ignore interactions between LOF mutations on the same background, or on the other chromosome within the same gene. Moreover,

our model does not apply to genes that are fully recessive in their fitness effects (as distinct from their phenotypic effects); we expect such genes to be rare (*Amorim et al., 2017*). A more subtle, although standard (e.g., *Cassa et al., 2017*; *Dukler et al., 2022*; *Sawyer and Hartl, 1992*; *Simons et al., 2014*; *Weghorn et al., 2019*; *Williamson et al., 2005*), choice is that *hs* is modeled as fixed through time, even as the environment fluctuates and as the effective population size changes dramatically. Our observations of a strong enrichment of highly deleterious mutations in severe disease cohorts suggest that strongly selected mutations typically remained so to the present day, but that may not be the case for more weakly deleterious mutations. Finally, the parameter *hs* can be conceptualized as the product of its average fitness cost in individuals where it has an effect, and its penetrance in the population with regard to the various phenotypes to which it contributes. Thus, while – again as is standard – we modeled *hs* as fixed in all carriers (e.g., at 10%), it may instead be worth considering allele dynamics if *hs* varied among carriers (e.g., were 1 in 10% of carriers). Regardless, these aspects can readily be addressed within the same framework, in extensions of this work.

Another challenge in estimating *hs* – as well as proxies such as measures of 'mutation intolerance' – arises from more general difficulty of generalizing from biomedical samples that were collected with various ascertainment biases. One concern is that the health of these samples is non-representative of the general population. For gnomAD specifically, although individuals known to be affected by severe pediatric disease and their first degree relatives were removed, there are nonetheless some individuals who are cases ascertained for disease (*Gudmundsson et al., 2022*; *Karczewski et al., 2020*); if the allele frequencies of some LOF mutations are elevated because of this ascertainment, we would underestimate the fitness costs for those genes.

Despite these limitations, our estimates of *hs* seem sensible in a number of respects. As expected from first principles, they suggest stronger selection on the loss of a gene copy on the X than the autosomes, other than in the PAR. They are on average higher for DNMs and very rare segregating variants than variants at high allele frequencies in the population. And they reveal an enrichment of strongly deleterious mutations in cases for early-onset disorders in rough accordance with their severity.

Moreover, anchoring observations in human genetics in a population genetic model allows different phenotypes to be viewed within a shared framework through their relationship to fitness. In doing so, it helps to characterize the mutations mapped to date in different disease studies: how likely they are to be causal, how many generations they are likely to persist, and at what frequencies we should typically expect to see them in other populations.

A further nice feature of interpreting findings of mapping studies in terms of DFEs is that it provides a way to characterize and compare the deleterious effects of variants found in different types of disease cohorts, potentially helping to guide discovery of causal variants (*Chakravarti and Turner, 2016*). As an illustration, for autism, our analysis indicates that a cohort of affected females in simplex families should yield many more highly deleterious causal variants than a mixed cohort of similar size. It further implies that comparing largely male cases to female controls may substantially reduce power to detect causal mutations (as unaffected females may harbor incompletely penetrant mutations). Additionally, our findings suggest that simplex family designs might provide the greatest insight into large effect causal mutations on low liability backgrounds. In turn, since large families with many affected individuals rarely seem to harbor germline mosaic mutations transmitted to multiple offspring, or independent causal DNMs in multiple offspring, they may instead be most informative about high-risk polygenic backgrounds and causal mutations of smaller effects.

Moving forward, estimates of fitness costs such as the ones reported here for LOF mutations can also be obtained for missense and regulatory mutations, indels, and CNVs (*Agarwal and Przeworski, 2021*; *Chen et al., 2022*; *Dukler et al., 2022*; *Halldorsson et al., 2021*; *Smolen and Girirajan, 2022*; *Zhang et al., 2022*). In addition to helping to prioritize variants, such estimates will allow pathogenic effects of different mutation types to be compared, as well as aid in the interpretation of GWAS findings (e.g., *Grotzinger et al., 2022*; *Mostafavi et al., 2022*; *Sella and Barton, 2019*).

## Materials and methods

We inferred the strength of selection acting on the LOF of each gene. To this end, we compared the frequency of LOF variants expected given a plausible demographic model and mutation rate to the observed frequency of such variants in extant individuals (see *Figure 1* for a schematic). Below, we first

describe how observed data are obtained and processed from gnomAD (*Karczewski et al., 2020*), followed by an outline of our model and the inference scheme.

## Estimating hs

### Mutation rates

As in previous studies (*Cassa et al., 2017*; *Weghorn et al., 2019*), we made the simplifying assumption that after some filtering (see below), all LOF mutations in a gene have identical selection coefficients and thus each gene can be modeled as a single biallelic locus with a single mutation rate μ. Values of μ for each gene were obtained from the 'high-confidence' LOF mutation rates for autosomes and the X chromosome provided as part of the gnomAD 2.1.1 release. The underlying methodology is detailed in *Karczewski et al., 2020*. We excluded 507 genes that had μ = 0, i.e., did not have a (known) mutation rate to LOF.

We checked the validity of the gnomAD mutation model by gauging its fit to DNM data for the X chromosome and autosomes (*Appendix 1—figure 1*). To this end, we categorized autosomal genes by quartiles of the mutation rate estimates $\mu_{total}$ (over synonymous, missense, and LOF sites in a gene) from gnomAD. We summed $\mu_{total}$ over all genes within each quartile and divided by $\mu_{total}$ over all genes in the exome to obtain the per-quartile haploid mutation rate for the gnomAD mutation model. For comparison, we calculated the DNM rate in each group of genes: exonic DNMs on the X and autosomes were obtained from the DDD (*Kaplanis et al., 2020*) and Decode studies (*Halldorsson et al., 2019*; *Jónsson et al., 2017*). Although the individuals in the former study, and some in the latter, were ascertained for severe disease, and there may be some expected enrichment of LOF mutations as a result, the exonic mutation rate in these studies is comparable. We also used exonic DNMs from *Goldmann et al., 2016* that are not ascertained on a disease phenotype, and similarly comparable to the ascertained sets in the overall mutation rate; however, no data for the X were available. We obtained 95% Poisson confidence intervals for the DNM counts in each quartile. Because of much smaller amounts of DNM data for the X chromosome, we categorized X chromosome genes into two groups instead of four.

### Observed frequency of LOF variants

We downloaded whole-exome polymorphism data for 141,456 individuals made available as part of gnomAD 2.1.1 (*Karczewski et al., 2020*). These data are polarized to the reference genome (hg19) and annotated with variant consequences using Variant Effect Predictor (v85, Gencode V19) and the LOFTEE tool to flag high-confidence ('HC') LOF variants.

We excluded genes with duplicate IDs or conflicting names between Gencode and gnomAD (n = 46). We excluded a variant if (i) it did not pass quality control in gnomAD (using the 'Filter' column in the vcf files); (ii) it was an indel; (iii) it was not 'high-confidence' LOF, per the criteria enumerated in *Karczewski et al., 2020*, in the canonical transcript of the gene, and (iv) if the total number of (reference and alternate) alleles for the variant was lower than 2 standard deviations below the mean allele number in the NFE sample, calculated separately for autosomes and the PAR, and the non-PAR X. We then summed the allele frequencies of the remaining variants within each gene in the NFE sample of 56,855 individuals to obtain the observed frequency of LOF mutations per gene. We excluded 793 genes for which fewer than 50% of 'high-confidence' LOF mutations met the above threshold on allele number.

### Forward simulations on autosomes and the pseudoautosomal region

To model LOF mutations in a gene, we used a forward population genetic simulation framework initially described in *Simons et al., 2014*, and adapted for LOF mutations in *Fuller et al., 2019*. Briefly, a gene is modeled as a single non-recombining biallelic locus that undergoes mutation to an LOF allele each generation at rate 2Nμ in a panmictic diploid population of size N; we further assume new mutations can arise only on a background free of other LOF variants and that back mutations occur at a rate of 0.01μ. Assuming identical fitness effects for all LOF mutations in a gene as described in the 'Mutation rates' section, compound heterozygotes implicitly have the same fitness effect as homozygotes. We assume that mutations are not fully recessive, where fully recessive is defined as *2Nhs* << 1 and *2Ns* >> 1.

Given μ for the gene of interest and an appropriate demographic model, we simulate the evolution of this locus forward in time under a single dominance coefficient (*h*) and selection coefficient (*s*) to obtain the frequency of LOF at present (i.e., the sum of the frequencies of any LOF alleles in the gene). Each generation is formed by Wright–Fisher sampling with selection, with parents chosen according to their fitness. As a starting demographic model, we use the Schiffels–Durbin model for population size changes in Europe over the past ~55,000 years (*Schiffels and Durbin, 2014*), preceded by an ~10N generation burn-in period of neutral evolution at an initial population size *N* of 14,448 (following *Simons et al., 2014*; *Simons et al., 2018*). In the last generation, i.e., at present, we sample 2n chromosomes from the simulated population, to match the size of the NFE samples with good coverage for the gene in gnomAD. The simulations are implemented in C++ and available online at https://github.com/zfuller5280/MutationSelection (*Agarwal, 2023* copy archived at swh:1:rev:847d659a71a0f8bd04bcd68fa26a18b0b99ad255).

## Forward simulations on the non-PAR X chromosome

On the autosomes, we do not need to model the two sexes separately, and all parameters can be specified as averages across sexes. In contrast, on the X chromosome (outside of the PAR), we need to incorporate sex-specific mutation rates, mating with two sexes, and different modes of selection in males and females (because males are hemizygous for the X chromosome, and because there is X-inactivation in females). To this end, we alter the above simulation framework in the following ways.

First, we introduce mutations in males at the rate $\mu_m$ and in females at the rate $\mu_f$, where the sex-specific mutation rates can be expressed in terms of the sex-averaged mutation rate μ on the X chromosome, and α, the ratio of the male mutation rate to the female mutation rate, as follows:

$$\mu_m = 3\mu * \alpha/(2 + \alpha)$$
$$\mu_f = 3\mu * 1/(2 + \alpha)$$

The sex-averaged LOF mutation rate for genes on the X are obtained as for genes on the autosomes (see the 'Mutation rates' section above). Unless otherwise specified, we use an α of 3.5 (see Figure 2B in *Gao et al., 2019*). Although there could potentially be differences in the male bias in mutation rate across genes (e.g., due to sex differences in transcription rates and replication timing), in practice these effects are expected to be small (*Aggarwala and Voight, 2016*; *Seplyarskiy and Sunyaev, 2021*).

We note that the total number of mutations every generation on the X chromosome is *3Nμ* on average, regardless of the value of α, but with large values of α, more mutations enter the population through males on average, even though the number of X chromosomes is twice as high in females.

Second, mating occurs between two parents of the opposite sex. We separately track male and female offspring born in each generation (with a fixed sex ratio of 0.5). We implicitly assume that there is no sex difference in demographic history and that the variance in reproductive success is the same for the two sexes.

Third, on autosomes, female heterozygotes for an LOF allele experience a fitness cost *hs*, and homozygotes *s*. On the X, in males, the fitness cost of the loss of the only copy of the gene is *s*. Female heterozygotes and homozygotes for LOF alleles on the X experience a fitness cost of *hs* and *s*, respectively, although the dominance coefficient in female heterozygotes has a slightly different interpretation for genes that undergo X-inactivation. We verified that our model of mutation and fitness on the X chromosome matched expectations under mutation-selection balance in a constant population size under a range of selection coefficients (*Appendix 1—figure 6*).

As is standard, selection in our Wright–Fisher implementation implicitly operates on fertility (i.e., in the parental generation) and not on viability of embryos. Under the simplifying assumption that selection pressures are the same in gametogenic and embryonic stages, this implementation correctly proxies viability selection on autosomes. Viability and fertility selection cannot be treated as equivalent on the X chromosome, however, because the X chromosome is passed from father to daughter and from mother to son, and the mode of selection is different in the two sexes. New mutations arising on the X in the female germline experience fertility selection in the heterozygous state, then viability selection in the hemizygous state in the male offspring; mutations arising on the X in the male germline undergo fertility selection in hemizygous males followed by viability selection in a female embryo in the heterozygous state. Thus, under the standard implementation, newly arising mutations on the X

in males would experience on average more selection than they would under a model of true viability selection. To better approximate viability selection on newly arising mutations on the X, we altered our implementation such that mutations arising on the X in the male germline undergo selection in the heterozygous state, as they would in female embryos, and mutations arising on the X in the female germline undergo hemizygous selection, as they would on the X chromosome in a male embryo. This is expected to have only a small effect; we verified that it makes no discernible difference to the results (see *Appendix 1—figure 7*).

## Testing the demographic model in forward simulations

The Schiffels–Durbin model includes an $N_e$ of 613,285 over the last 124 generations (*Schiffels and Durbin, 2014*). To assess how well this period of recent population growth explains variation in the observed data, we compared two different measures of neutral polymorphism in the NFE sample to simulations with hs = 0: (i) the proportion of segregating synonymous sites in each gene and (ii) following *Weghorn et al., 2019*, the frequency spectrum of all synonymous non-CpG transversions, a mutation type that occurs at low rate and thus should include few multiple hits at a site. Specifically, for modeling synonymous variants in each gene, we took the per gene synonymous mutation rate reported by gnomAD and divided by the total number of synonymous mutational opportunities to obtain a mean per site mutation rate μ for the forward simulations described above. Simulating under hs = 0, we then generated the expected proportion of segregating sites for each gene. For modeling non-CpG transversions, we used $\mu = 3.8 \times 10^{-9}$ (*Kong et al., 2012*; *Weghorn et al., 2019*) and compared the simulated frequency spectrum from $10^6$ simulations under hs = 0 to the observed spectrum for all non-CpG synonymous transversions in the NFE sample. The standard Schiffels–Durbin demographic model underestimated the proportion of segregating synonymous sites in simulations for nearly all genes on both the autosomes and X chromosome (*Appendix 1—figure 2*). Moreover, the simulated frequency spectrum was shifted away from rare variants relative to the observed data and the fraction of singletons was substantially lower in simulations (0.373) than in the NFE sample (0.637) (*Appendix 1—figure 3*).

We therefore modified the Schiffels–Durbin model to include an additional epoch of growth over the last 50 generations with an $N_e$ of 5 million and again compared measures of neutral polymorphism between simulations and observed data in the NFE sample. Using this modified demographic model, we observed improved agreement between the proportion of segregating synonymous sites in simulations and the observed data for autosomal and X-linked genes (*Appendix 1—figure 2*). Additionally, the frequency spectrum for synonymous non-CpG transversions appeared more similar and the fraction of singletons in simulations (0.677) more closely matched that of the NFE sample (*Appendix 1—figure 3*). Thus, for all subsequent analyses and simulations, we relied on this modified Schiffels–Durbin demographic model.

## Expected frequency of LOF variants under neutrality

We first obtained the expected frequency of LOF variants in each gene under neutrality (i.e., hs = 0). For each per gene LOF mutation rate μ, we performed 50,000 simulations and estimated where the observed LOF frequency in the NFE sample fell within the resulting distribution. Genes where the observed LOF frequency was ≥90% of the simulated frequencies under neutrality were classified as cases where our model of purifying selection is misspecified. We note there are several, nonmutually exclusive, alternative explanations for cases where the observed LOF frequency greatly exceeds that of the neutral expectation, including an incorrect mutational model, balancing selection, annotation errors. In total, we classified 285 such genes (*Supplementary file 1*). These were removed from further analysis.

## Selection parameter (hs) inference

We estimated the posterior distribution of *hs* given the LOF allele frequency and mutation rate μ for a gene under a sensible demographic model for the NFE population.

To estimate *hs*, we used an Approximate Bayesian Computation (ABC) approach, which consists of three basic steps: (i) proposing parameters from a prior distribution, (ii) simulating data under a generative model using the proposed parameters, and (iii) retaining parameters that closely match the observed data, within some tolerance. Specifically, for each iteration *i* in our ABC implementation,

we proposed a value of *hs* for autosomes by sampling from $log_{10}\left(s\right) \sim U\left(-6, 0\right)$, and $h \sim U\left(0, 1\right)$; for the X chromosome, we proposed *hs* for females and *s* for males, with *h* and *s* sampled separately as above (as a result, we have the same prior on *hs* for the X and autosomes). We then generated an allele frequency $q_i$ using the forward simulations described above. This simulated allele frequency is compared to the observed allele frequency *q* in gnomAD data for the gene, and accepted if $|q_i\text{-}q|$ < ε, where ε is the tolerance. When $|q_i\text{-}q|$ = 0, the retained parameters are a sample from the posterior distribution of *hs* given the allele frequency of LOF mutations in the gene. For small ε values, however, the acceptance rates can become too low, thus making ABC computationally inefficient. To alleviate this issue, we used an ABC based on a Sequential Monte Carlo algorithm (ABC-SMC), with the idea of gradually moving from sampling the entire prior for proposal values to sampling from the target posterior distribution, through a sequence of intermediary distributions based on a decreasing schedule of ε values (*Sisson et al., 2007*). We implemented an ABC-SMC approach using the modular C++ library 'pakman' (*Pak et al., 2020*) and set a tolerance schedule for allele frequencies as ε = $\left\{\frac{10}{2n}, \frac{5}{2n}, \frac{2}{2n}, 0\right\}$. At each ε, we obtained 50,000 samples from the distribution. We report per gene point estimates of *hs* obtained from the MAP estimate of the posterior and uncertainty measured by the 95% credible interval (CI) (*Supplementary file 2*). For the X chromosome outside the PAR, we report the sex-averaged strength of selection on the loss of a copy, calculated as (*hs* + *s*)/2.

We verified the reliability of our ABC-SMC approach by simulations under a range of selection coefficients, comparing it to the true posterior distribution and to the posterior distribution inferred using rejection-ABC for 50,000 samples with $\varepsilon$ = 0 for simulated genes (*Appendix 1—figure 4*).

## Estimating the age of LOF alleles segregating at present

We modified the forward simulations described above so that at most one mutation could arise each generation and only if the site is not segregating. We simulated evolution forward in time at an autosomal locus as above, under the same demographic model, and hs = 1%, 10%, or 50%. In each simulation, conditional on the site segregating at present, we recorded the last generation in which the locus was invariant in the population, and thus obtained the distribution of the age of an allele sampled in the population at present.

# Analyzing the fitness effects of possible and observed LOF mutations

## Data sources and processing

### Mutational opportunities on the X and autosome

We obtained the total number of possible 'high-confidence' LOF mutations for each gene on the X and Autosome provided as part of the gnomAD 2.1.1 release (*Karczewski et al., 2020*).

### De novo mutations in unaffected individuals

We obtained publicly available DNMs in a hospital cohort of ~800 newborns not ascertained for any disease (*Goldmann et al., 2016*). We annotated variants using Variant Effect Predictor (v85, Gencode v19) and kept only exonic variants.

Similarly, we obtained DNMs in ~1800 unaffected siblings in the Simon Simplex autism study (*An et al., 2018*). We lifted the variants over to the hg19 assembly and annotated the variants using Variant Effect Predictor (v85, Gencode v19), and kept variants classified as LOF.

In addition to DNMs seen in surviving offspring, we also downloaded mutations seen in spermatogonial stem cells from 13 individuals (*Moore et al., 2021*). Mutations were pre-annotated; we retained those labeled as LOF.

Only autosomal variants were available for all three sources.

### Segregating variants in the population

We downloaded the population-level plink files with exome-wide genotype information for ~200,000 individuals released by the UK Biobank (*Szustakowski et al., 2020*). We excluded exome samples that did not pass variant or sample quality control criteria in the previously released genotyping array data. Specifically, we excluded samples that have a discrepancy between reported sex and inferred sex from genotype data, a large number of close relatives in the database, or are outliers based on heterozygosity and missing rate, as detailed in *Bycroft et al., 2018*. We excluded individuals who

withdrew from the UK Biobank by the time of analysis. This left us with 199,930 individuals that are included among the high-quality subset of genotyped individuals. We additionally limited our analysis to the ~166K individuals designated as 'White British' in the original study, and to the list of ~38 million exonic sites with an average of 20× sequence coverage provided by UK Biobank, for which variants met the QC criteria described in *Szustakowski et al., 2020*. We excluded the small subset of variants for which the number of homozygotes and heterozygotes are not consistent with Hardy–Weinberg proportions (p-value cutoffs of ~$10^{-5}$ vs. $10^{-2}$ made no difference to the results).

We transformed the processed plink files into the standard variant call format, polarized variants to the hg38 reference assembly (i.e., the reference allele is considered ancestral), and lifted over the coordinates from hg38 to hg19 using the UCSC LiftOver tool. The few positions where the reference alleles were mismatched or swapped between the two assemblies were excluded. We annotated the ~9 million variants with variant consequences using Variant effect predictor (v85, Gencode V19) and the hg19 LOFTEE tool to flag high-confidence ('HC') LOF variants. We then used these annotations to exclude all variants that are not 'high-confidence' LOF in the canonical transcript. Where there are multiple canonical transcripts or multiple consequences per canonical transcript, we picked the variant with the most deleterious consequences using ranks provided by ensembl since those are the criteria used by many studies that map mutations in disease.

For each individual in this sample, we also obtained a list of all genes with heterozygous LOF. In counting LOF variants per individual, we considered variants that overlap two genes to result in an LOF in both (alternatively, we could choose one at random; in practice, the choice makes little difference to the counts).

We also obtained the above information for the subset of 110,667 individuals who self-report no long-standing illness, disability or infirmity (Field ID 2188) in the UK Biobank.

## De novo mutations and rare segregating variants mapped in severe complex diseases

We obtained published DNMs from various sources. For each study, we retained only LOF mutations (annotated as 'stop-gained,' 'splice donor,' 'splice acceptor,' 'esplice,' 'nonsense,' or 'LGD'). For the MSSNG dataset, we annotated mutations using Variant Effect Predictor (v85, Gencode v19). Where available, we also retained information about the siblings, disease status of family members, age of onset, and age and sex of probands.

We focused on six disorders for which substantial numbers of DNMs were publicly available: DD; CHD; developmental and epileptic encephalopathies; autism; schizophrenia; and Tourette's syndrome or OCD (*Cappi et al., 2020*; *EuroEPINOMICS-RES Consortium et al., 2014*; *Fromer et al., 2014*; *Hamdan et al., 2017*; *Howrigan et al., 2020*; *Jin et al., 2017*; *Kaplanis et al., 2020*; *Rees et al., 2020*; *Satterstrom et al., 2020*; *Willsey et al., 2017*; *Xu et al., 2012*). We combined the DNM lists for Tourette's syndrome and OCD because a large fraction of individuals in the two groups were diagnosed with both conditions (*Cappi et al., 2020*; *Willsey et al., 2017*).

We used the pedigrees from *Satterstrom et al., 2020*, which contained individuals from the Simons Simplex study (SSC), and the Autism Sequencing Consortium (ASC), for our analysis of mutations underlying autism. Because the ASC in particular draws samples from a wide variety of cohorts for which we did not have study-specific information, we used three nonoverlapping cohorts (Simons Simplex, SPARK, and MSSNG; *C Yuen et al., 2017*; *Feliciano et al., 2019*; *Fischbach and Lord, 2010*) that differ in known ways with regard to their composition, to investigate the effects of cohort composition on the DFE. The Simons Simplex data are ascertained to be enriched for simplex families. To reduce the likelihood of multiplex families being misclassified as simplex (e.g., possible if the parents only have one child, if siblings were too young at diagnosis, if siblings have a milder phenotype, etc.), all probands in the Simons Simplex cohort have at least one sibling ascertained to not meet the diagnostic criteria for autism, in addition to unaffected parents (*Fischbach and Lord, 2010*). The MSSNG data contain both simplex and multiplex families: we classified affected individuals as belonging to multiplex families if they had at least one affected sibling reported, and as simplex if they had no affected family members (*C Yuen et al., 2017*). For the SPARK study, we did not have information on which individuals are in simplex vs. multiplex families, only the overall cohort composition: 418 simplex and 39 multiplex families (with 47 affected individuals) (*Feliciano et al., 2019*).

For schizophrenia, we combined DNMs from four samples (*Fromer et al., 2014*; *Howrigan et al., 2020*; *Rees et al., 2020*; *Xu et al., 2012*), including one of Taiwanese individuals (*Howrigan et al., 2020*). We verified that combining the European samples and the samples from Taiwan did not affect our conclusions (*Appendix 1—figure 13*). Since we did not have information about simplex and multiplex families, or affected siblings, we used the presence of reported family history of schizophrenia or other mental illness as a proxy for multiplex families (*Appendix 1—figure 13*).

We also downloaded rare segregating variants in cases and controls, available publicly for epilepsy, autism, schizophrenia, and bipolar disorder (*Feng et al., 2019*; *Palmer et al., 2022*; *Satterstrom et al., 2020*; *Singh et al., 2022*). Note that each study defined rare variants based on their own criteria: for example, the autism study designates rare variants as those with 'allele frequency $\leq 0.1\%$ in our dataset and non-psychiatric subsets of reference databases' (*Satterstrom et al., 2020*).

Data sources are summarized in *Appendix 1—table 2*.

## Obtaining the DFE from hs estimates

Using the inferred posterior distributions of *hs* for the LOF of each gene, we obtained the DFE for all possible de novo LOF mutations in the genome by weighting the posterior for each gene by its contribution to genome-wide mutational opportunities to an LOF allele. Consistent with our modeling assumption, all possible LOF mutations within the same gene are assumed to have the same posterior distribution of *hs*. Similarly, the DFE for any sample of LOF mutations is obtained by weighing the posterior density of *hs* for each gene with the fraction of observed LOF mutations in the gene.

## Comparing DFEs of observed mutations to those expected by chance

For *n* DNMs in a disease cohort, we bootstrapped 1000 DFEs of a set of *n* DNMs randomly sampled (with replacement) from the full set of LOF mutational opportunities in the genome. In other words, each bootstrapped distribution is a sample from the distribution of fitness effects over all possible LOF mutational opportunities in the genome. p-Values were calculated using the rank of the mean of the distribution for each disease compared to the means of the 1000 bootstrapped distributions.

## Comparing DFEs of disease mutations for enrichment of highly deleterious mutations

We bootstrapped 500 samples each from the two DFEs with replacement and calculated the area in the interval (0.1,1) in each sampled DFE. We compared the distributions of sampled areas for the two diseases; p-values were obtained from a Kolmogorov–Smirnov test.

## Calculating the probability of being causal

The probability of hs > 10% is calculated as the area under the DFE in the interval (0.1,1). The probability that a mutations with hs > 10% is causal for a disease is calculated as

$$1 - \left( P\left( hs > 10\% \text{ in a population sample} \right) / P\left( hs > 10\% \text{ in the disease cohort} \right) \right)$$

# Acknowledgements

We thank Peter Andolfatto, Jeremy Berg, Arbel Harpak, Kelley Harris, Edith Heard, Hakhamanesh Mostafavi, Magnus Nordborg, Itsik Pe'er, Jonathan Pritchard, Guy Sella, and members of the Andolfatto, Przeworski and Sella labs for helpful discussions, as well as Jonathan Pritchard and Guy Sella for comments on an earlier draft of the manuscript. We are grateful to Joanna Kaplanis for sharing DDD data and Konrad Karczewski for help with the gnomAD dataset and LOFTEE. This work was supported by NIH grants GM121372 and HG011432 to MP, NRSA GM128318 to ZF, and WT Investigator Award 212284/Z/18/Z to SRM.

# Additional information

### Competing interests

Molly Przeworski: Senior editor, *eLife*. The other authors declare that no competing interests exist.

## Funding

| Funder | Grant reference number | Author |
|---|---|---|
| National Institutes of Health | GM121372 | Molly Przeworski |
| National Institutes of Health | HG011432 | Molly Przeworski |
| National Institutes of Health | GM128318 | Zachary L Fuller |
| Wellcome Trust | WT Investigator Award 212284/Z/18/Z | Simon R Myers |

The funders had no role in study design, data collection and interpretation, or the decision to submit the work for publication. For the purpose of Open Access, the authors have applied a CC BY public copyright license to any Author Accepted Manuscript version arising from this submission.

## Author contributions

Ipsita Agarwal, Zachary L Fuller, Conceptualization, Data curation, Formal analysis, Investigation, Visualization, Methodology, Writing – original draft, Writing – review and editing; Simon R Myers, Methodology, Writing – review and editing; Molly Przeworski, Conceptualization, Resources, Supervision, Investigation, Methodology, Writing – original draft, Project administration, Writing – review and editing

## Author ORCIDs

Ipsita Agarwal http://orcid.org/0000-0001-8537-0008
Zachary L Fuller http://orcid.org/0000-0003-4765-9227
Molly Przeworski http://orcid.org/0000-0002-5369-9009

## Decision letter and Author response

Decision letter https://doi.org/10.7554/eLife.83172.sa1
Author response https://doi.org/10.7554/eLife.83172.sa2

# Additional files

## Supplementary files

• MDAR checklist

• Supplementary file 1. List of 285 putatively "neutral" genes.

• Supplementary file 2. Point estimates and confidence intervals for hs for 17,318 autosomal and 679 X chromosome genes.

## Data availability

All source data are freely available to researchers, with sources listed in *Appendix 1—table 2*. Code for simulations, and output is available at https://github.com/zfuller5280/MutationSelection (copy archived at swh:1:rev:847d659a71a0f8bd04bcd68fa26a18b0b99ad255) and https://github.com/agarwal-i/loss-of-function-fitness-effects (copy archived at swh:1:rev:ff59eb663346354e5d32ec-589ca3d6afddc705fb). Estimates of fitness costs of LOF mutations are provided as *Supplementary file 2*.

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

# Appendix 1

**Appendix 1—table 1.** Summary counts for LOF and synonymous mutations by pedigree study or subsample.

Studies can differ in the amount of the genome queried for LOF mutations. The number of probands is obtained as the number of unique Proband IDs with at least one reported de novo mutation (of any kind). For two studies where this information was not available, the number of probands was obtained from the text. The probability that an LOF mutation is under hs > 10% is calculated using the area under the DFE for hs > 10%. The probability that an LOF mutation under hs > 10% is causal is calculated as 1 – the ratio of the probability that an LOF mutation is under hs > 10% in a sample ascertained for a disease and the probability that an LOF mutation is under hs > 10% in the population (19.8%), and set to zero if negative.

| Sample | # Affected Individuals | Average number of Synonymous DNMs in an individual | Average number of LOF DNMs in an individual | Probability LOF has hs > 10% | Probability LOF causal if hs > 10% |
|---|---|---|---|---|---|
| Developmental disorders (2623 DNMs in 23,902 trios) | 23,902 | 0.38 | 0.11 | 0.50 | 0.60 |
| Congenital heart disease (192 DNMs in 1785 trios) | 1785 | 0.39 | 0.11 | 0.36 | 0.45 |
| Severe epilepsy (58 DNMs in 406 trios) | 406 | 0.35 | 0.14 | 0.46 | 0.57 |
| Autism (560 DNMs in 5297 trios) | 5297 | 0.35 | 0.11 | 0.35 | 0.43 |
| Schizophrenia (263 DNMs in 2381 trios) | 2381 | 0.24 | 0.11 | 0.28 | 0.28 |
| Tourette syndrome/OCD (62 DNMs in 436 trios) | 436 | 0.39 | 0.14 | 0.25 | 0.19 |
| DDD (445 DNMs in 4336 affected males) | 4336 | 0.41 | 0.10 | 0.49 | 0.59 |
| DDD (462 DNMs in 3411 affected females) | 3411 | 0.39 | 0.14 | 0.52 | 0.62 |
| Simons Simplex (117 DNMs in 1623 affected males) | 1623 | 0.25 | 0.07 | 0.35 | 0.43 |
| Simons Simplex (21 DNMs in 249 affected females) | 249 | 0.24 | 0.08 | 0.58 | 0.66 |
| MSSNG Simplex (38 DNMs in 531 affected males) | 531 | 0.32 | 0.07 | 0.40 | 0.51 |
| MSSNG Simplex (7 DNMs in 153 affected females) | 153 | 0.27 | 0.05 | 0.58 | 0.66 |
| SPARK (76 DNMs in 279 affected males) | 279 | 0.47 | 0.27 | 0.29 | 0.32 |
| SPARK (12 DNMs in 68 affected females) | 68 | 0.44 | 0.18 | 0.47 | 0.58 |
| MSSNG multiplex (35 DNMs in 491 affected males) | 491 | 0.36 | 0.07 | 0.19 | 0.00 |
| MSSNG multiplex (15 DNMs in 175 affected females) | 175 | 0.38 | 0.09 | 0.13 | 0.00 |

**Appendix 1—table 2.** Data sources by ascertainment.

| Ascertainment | Type | Study |
|---|---|---|
| Developmental disorders | DNMs | DDD; *Kaplanis et al., 2020* |
| Congenital heart disease | DNMs | *Jin et al., 2017* |
| Autism | DNMs | ASC and SSC whole-exome sequencing; *Satterstrom et al., 2020* |
| Autism (with unaffected sibling) | DNMs | SSC whole-genome sequencing; *An et al., 2018* |
| Autism | DNMs | SPARK; *Feliciano et al., 2019* |

*Appendix 1—table 2 Continued on next page*

*Appendix 1—table 2 Continued*

| Ascertainment | Type | Study |
|---|---|---|
| Autism | DNMs | MSSNG; *C Yuen et al., 2017* |
| Autism | Rare variants | *Satterstrom et al., 2020* (https://asc.broadinstitute.org/results) |
| Schizophrenia | DNMs | *Fromer et al., 2014*; *Howrigan et al., 2020*; *Rees et al., 2020* |
| Schizophrenia | Rare variants | *Singh et al., 2022* (https://schema.broadinstitute.org/) |
| Epilepsy | DNMs | *EuroEPINOMICS-RES Consortium et al., 2014*; *Hamdan et al., 2017* |
| Epilepsy | Rare variants | *Feng et al., 2019* (https://epi25.broadinstitute.org/) |
| Tourette's syndrome/OCD | DNMs | *Cappi et al., 2020*; *Willsey et al., 2017* |
| Bipolar disorder | Rare variants | *Palmer et al., 2022*, (https://bipex.broadinstitute.org/results) |
| Unknown | Segregating variants | *Szustakowski et al., 2020* UK Biobank Whole-exome sequences (https://biobank.ndph.ox.ac.uk/ukb/label.cgi?id=170) |
| Unknown | DNMs | *Goldmann et al., 2016* |
| Unknown | DNMs | Unaffected siblings in *An et al., 2018* |
| Unknown | Mutations in spermatogonial stem cells | *Moore et al., 2021* |
| Mixed | DNMs | *Halldorsson et al., 2019*; *Jónsson et al., 2017*; (the 2017 study contains DNMs on the X chromosome) |

**Appendix 1—figure 1.** Checking the gnomAD mutation model using de novo mutations (DNMs) in aggregate. Genes were grouped by quartiles of the gnomAD (*Karczewski et al., 2020*) mutation rate estimates $\mu_{total}$ (over synonymous, missense, and LOF sites in a gene), separately for (**A**) autosomes and (**B**) the X chromosome (only two groups were used for the X because of the very limited about of DNM data available for comparison). The per-quartile haploid mutation rate for the gnomAD mutation model was obtained by summing $\mu_{total}$ over all genes within each quartile and divided by $\mu_{total}$ over all genes in the exome. Exonic DNMs were obtained from DDD (*Kaplanis et al., 2020*), Decode (*Halldorsson et al., 2019*; *Jónsson et al., 2017*), and for autosomes only (since no data for the X were available) from *Goldmann et al., 2016*. We obtained 95% Poisson confidence intervals for the DNM counts in each quartile.

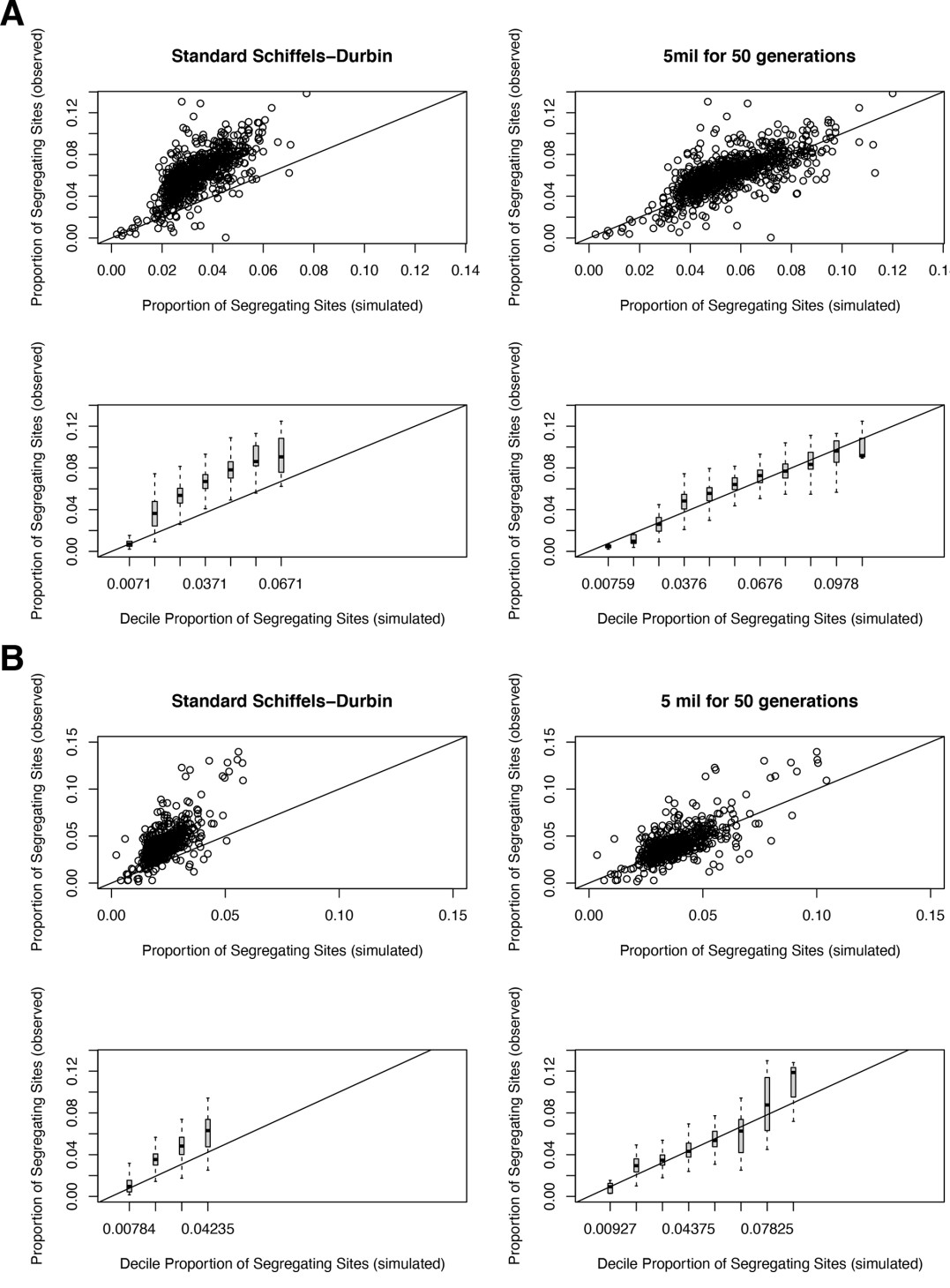

**Appendix 1—figure 2.** The proportion of segregating synonymous sites in genes for observed and simulated data. (**A**) For autosomal genes, the proportion of segregating synonymous sites in the gnomAD Non-Finnish European (NFE) sample on the y-axis compared to the proportion of segregating synonymous sites simulated under a neutral (hs = 0) model and per-gene synonymous mutation rates on the x-axis. Each point in the top row represents each gene, while the bottom row shows boxplots summarizing genes in increments of 0.01 segregating sites in simulations. On the left, genes are simulated under the widely used Schiffels–Durbin demographic model for population growth in Europe (**Schiffels and Durbin, 2014**). On the right, genes are simulated under a slightly modified version of this model, in which we set the effective population size Ne equal to 5 million for the past 50 generations. (**B**) The same plots, but for genes on the X chromosome.

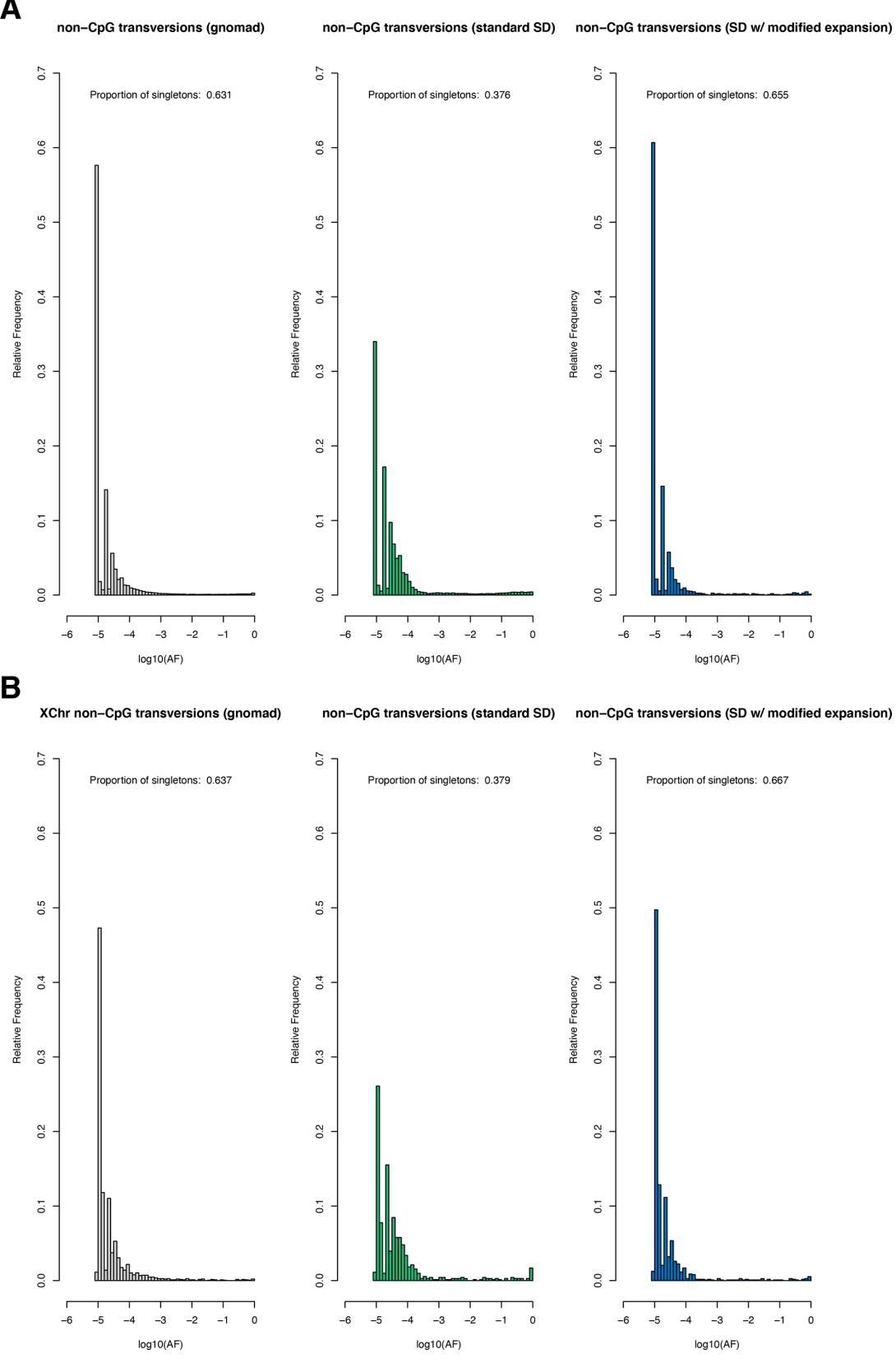

**Appendix 1—figure 3.** The site frequency spectrum for non-CpG transversions in observed and simulated data. (**A**) The frequency spectrum of synonymous non-CpG transversions for all autosomal sites in the Non-Finnish European (NFE) sample in gnomAD (left), simulated under the Schiffels–Durbin (*Schiffels and Durbin, 2014*) *Appendix 1—figure 3 continued on next page*

*Appendix 1—figure 3 continued*
demographic model for population growth in Europe ('standard SD'; middle), and simulated under a slightly modified version of this model in which we set the effective population size *Ne* equal to 5 million for the past 50 generations ('SD w/ modified expansion'; right). (**B**) The frequency spectrum of synonymous non-CpG transversions for all X chromosome sites. Columns are ordered the same as in (**A**).

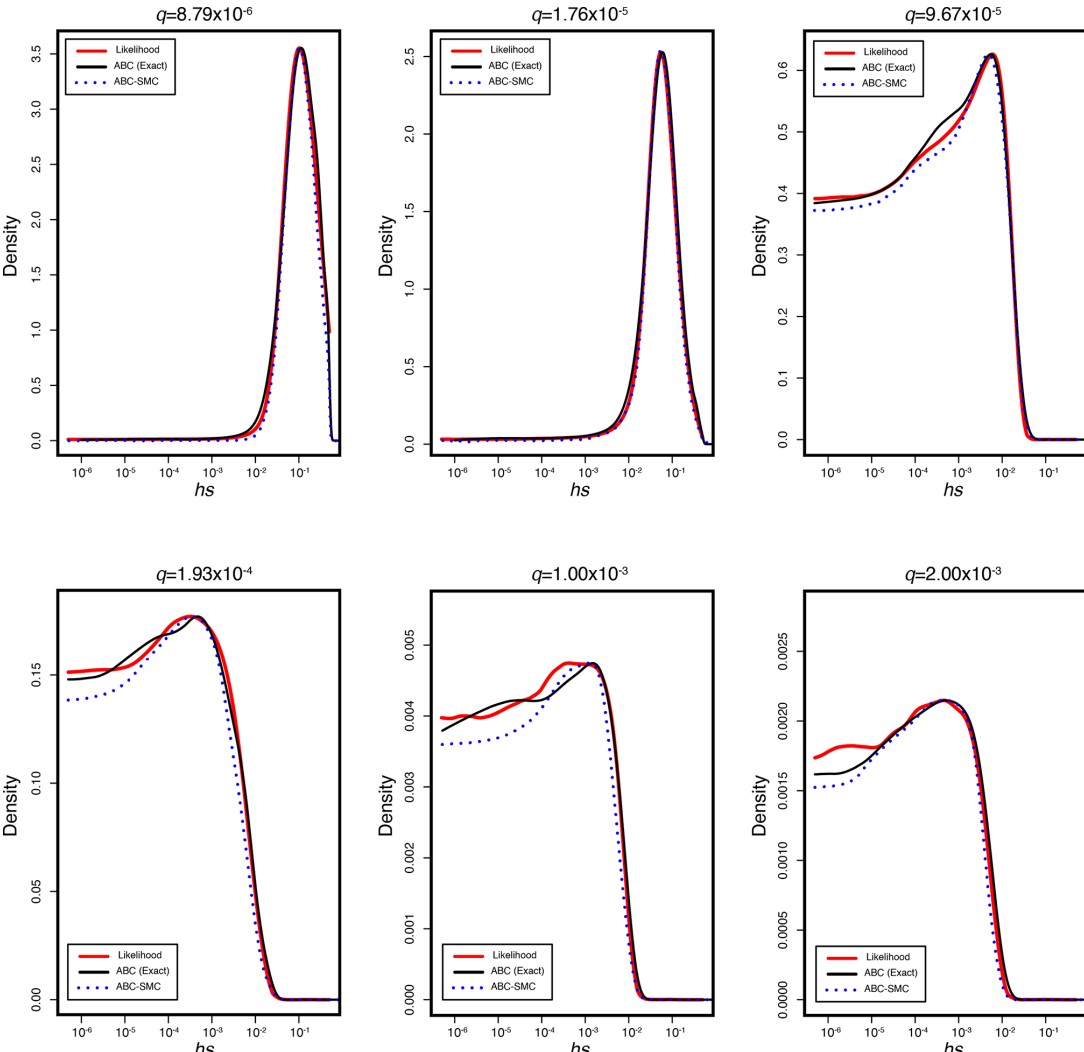

**Appendix 1—figure 4.** The true posterior and inferred posterior distributions of *hs* in simulated genes for six different observed loss-of-function frequencies *q*. In each, the red line represents the true posterior distribution of *hs* obtained from running $1 \times 10^6$ simulations for a gene with a mutation rate *u* to loss-of-function alleles of $1 \times 10^{-6}$ across a logarithmically spaced grid of 1000 *hs* values from $5 \times 10^{-6}$ to 1 under the modified Schiffels–Durbin demographic model described in the paper. The black curve represents the inferred posterior distribution using standard ABC rejection sampling with $\varepsilon = 0$. The dashed blue line indicates the inferred posterior distribution using the ABC-SMC approach described in the main text.

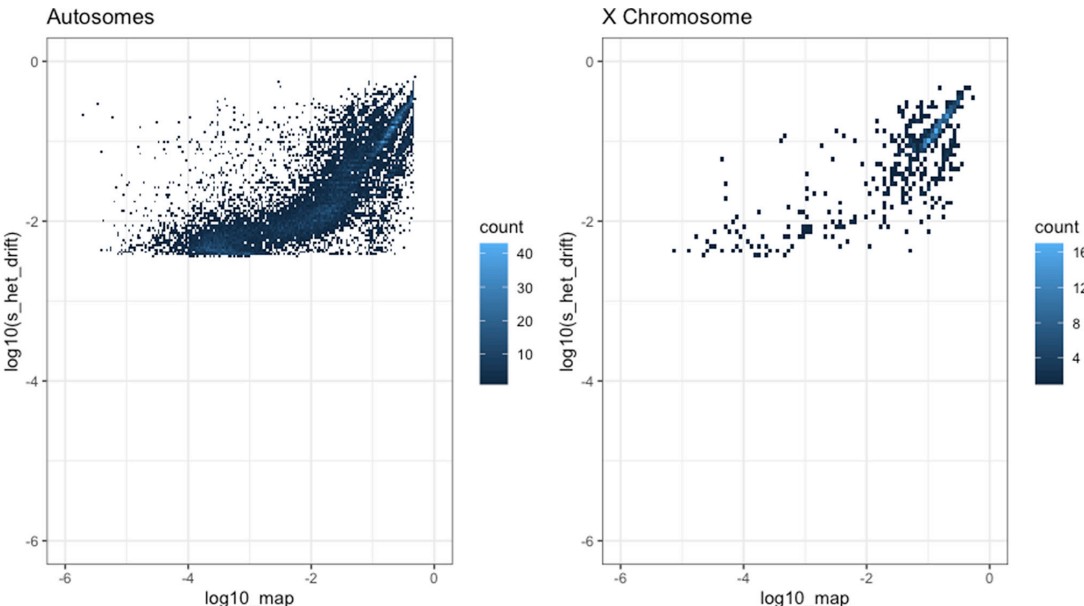

**Appendix 1—figure 5.** Comparison of our maximum a posteriori (MAP) estimates of *hs* (x-axis) and the *shet* measure from *Weghorn et al., 2019* (y-axis) for the autosomes (left) and X chromosome (right). Only the 15,275 genes for the autosomes and 583 genes for the X chromosome for which both estimates were available are included. For visualization, the color of each point reflects the number of overlapping genes on the plot.

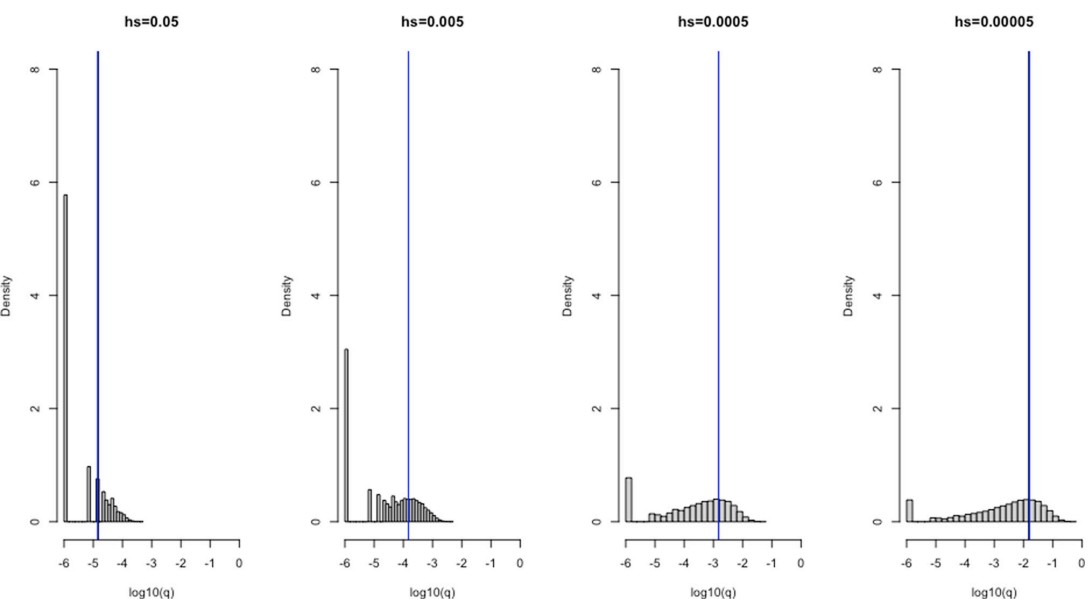

**Appendix 1—figure 6.** Verifying the model of selection on the X chromosome under a constant population size (N = 100,000). $1 \times 10^6$ simulations were run under four different heterozygous selection coefficients (*hs*), labeled above each plot. A mutation rate *u* of $1 \times 10^{-6}$ was set for all scenarios. The frequency *q* of variants for each scenario are shown on the x-axis on a $\log_{10}$ scale, and the density of the distribution is depicted on the y-axis. The blue line represents the mean allele frequency of the simulated distribution, and the black curve represents the expectation under mutation-selection-drift balance (calculated as *3 u/(2hs + s)*). The two are often visually indistinguishable.

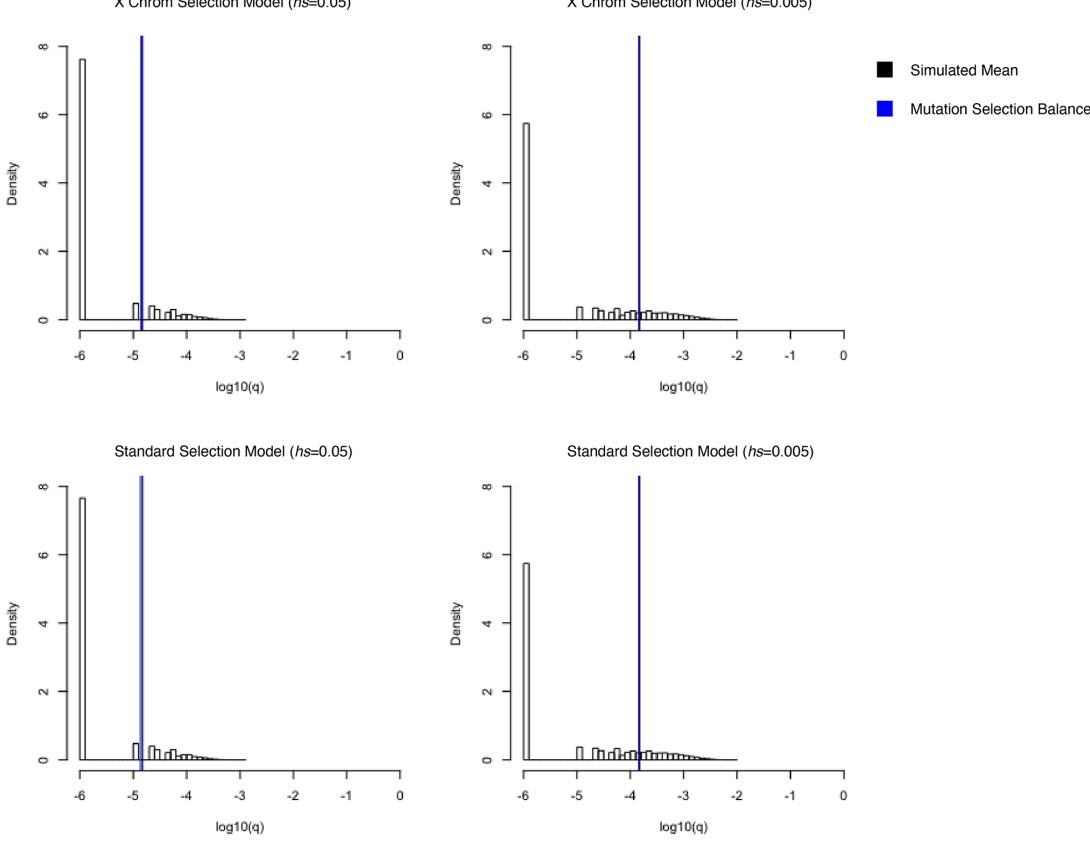

**Appendix 1—figure 7.** Simulated allele frequencies under hs = 0.05 (left) and hs = 0.005 (right) on the X chromosome for different models of selection. On the top row, simulations were performed using a model where mutations arising on the X in the male germline undergo selection in the heterozygous state and mutations arising on the X in the female germline undergo hemizygous selection. On the bottom row, simulations were performed using the standard model of viability selection, analogous to the one implemented on the autosomes. The vertical black curves represent the mean of 50,000 simulations, and vertical blue lines represent the expectation under mutation selection balance. The two are often visually indistinguishable.

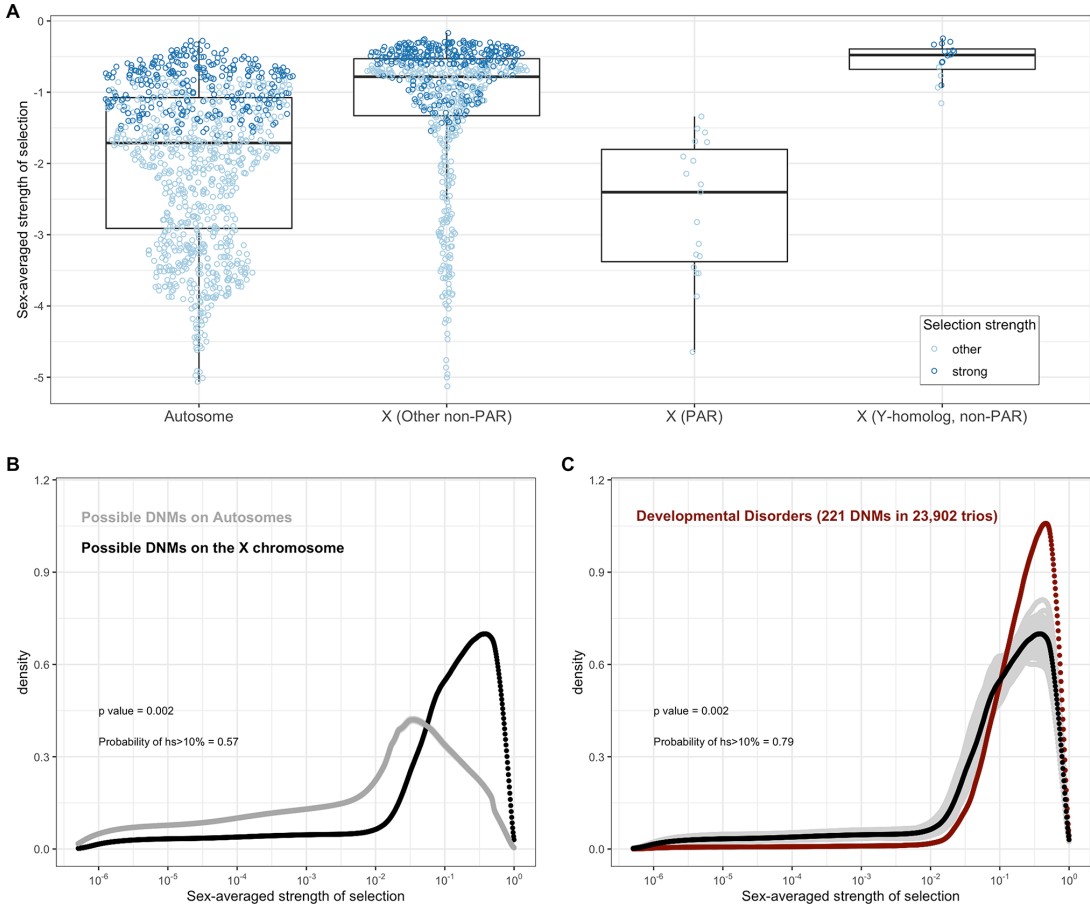

**Appendix 1—figure 8.** Estimated sex-averaged strength of selection on the loss of one gene copy on the autosomes and X. (**A**) The distributions of point estimates for the sex-averaged strength of selection on the loss of one gene copy on the autosomes and for three X chromosome compartments: the PAR (n = 19 genes), the non-PAR X without Y-homologs (n = 644), and the non-PAR X with Y homologs (n = 16). The PAR estimates are obtained using the autosomal model for inheritance, and the non-PAR X compartments under the X chromosome model. We also checked the estimates for the non-PAR X with Y homologs under an autosomal model (see 'Materials and methods'). For purposes of visualization, the dots for the autosomes represent a random sampling of 1000 genes. Lines represent the median values within each gene category. Each dot represents the point estimate for a single gene and is colored darker blue if 95% of the probability mass of the posterior distribution is greater than $hs > 10^{-2}$ (strong selection) and if not, lighter blue (other). (**B**) The estimated distribution of fitness effects (DFE) of all possible de novo loss-of-function (LOF) mutations on autosomes (in gray) compared to all possible de novo LOF mutations on the non-PAR X chromosome. 57% of the area under the DFE for X chromosome mutational opportunities corresponds to a selection strength of 10% or greater compared to 20% for autosomes. (**C**) The estimated DFE of de novo LOF mutations on the X chromosome in the Deciphering Developmental Disorders (DDD) cohort compared to the DFE of all mutational opportunities on the X (black curve), and 100 bootstrapped DFEs of a set of 221 de novo mutations (DNMs) randomly sampled from the full set of X chromosome LOF mutational opportunities (in gray). 79% of the area under the DFE for LOF mutations in the DDD cohort corresponds to a selection strength of 10% or greater compared to 57% of all LOF mutational opportunities on the X. We did not have sufficient data for the X for other disease cohorts.

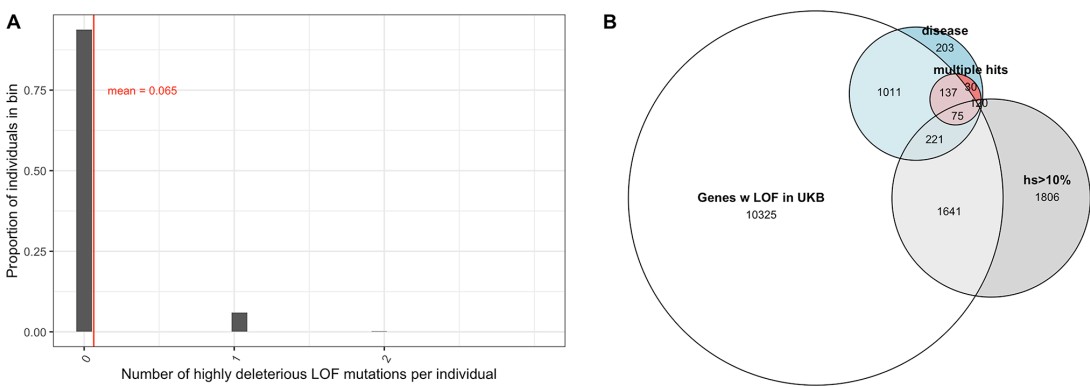

**Appendix 1—figure 9.** Variable penetrance of highly deleterious mutations.
 (**A**) The distribution of the number of segregating variants carried by individuals in the UK Biobank in genes with estimated hs >10% for loss-of-function (LOF). 93% of individuals carry no such LOF mutations, 6% of individuals carry one mutation, and 0.2% of individuals carry two. (**B**) Overlap of genes that have at least one LOF mutation segregating in the UK Biobank (among ~110K individuals who self-report no long-standing illness, disability, or infirmity; see 'Materials and methods') with (in gray) genes with estimated hs > 10% for LOF (in blue) genes with de novo mutations in individuals ascertained on severe disease (developmental disorders, congenital heart disease, autism, and epilepsy) and (in red) genes with DNMs mapped in at least two affected individuals, with at least one disease.

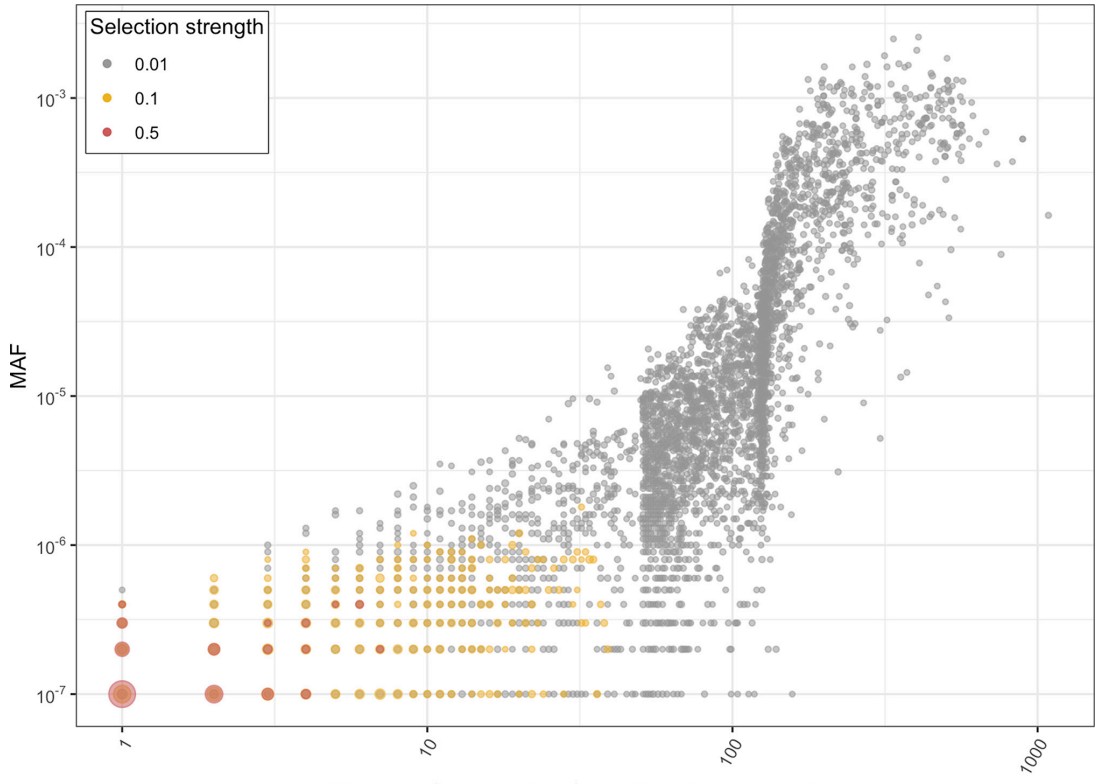

**Appendix 1—figure 10.** The distribution of the ages (in generations) and allele frequencies of a strongly selected loss-of-function (LOF) allele segregating in the population at present. Allele ages and allele frequencies in the population at present were obtained from 10,000 forward simulations at an autosomal locus under the modified Schiffels–Durbin demographic model described in the paper (see 'Materials and methods') and *hs* of 1%, 10%, and 50%. The size of a dot reflects the number of overlapping points.

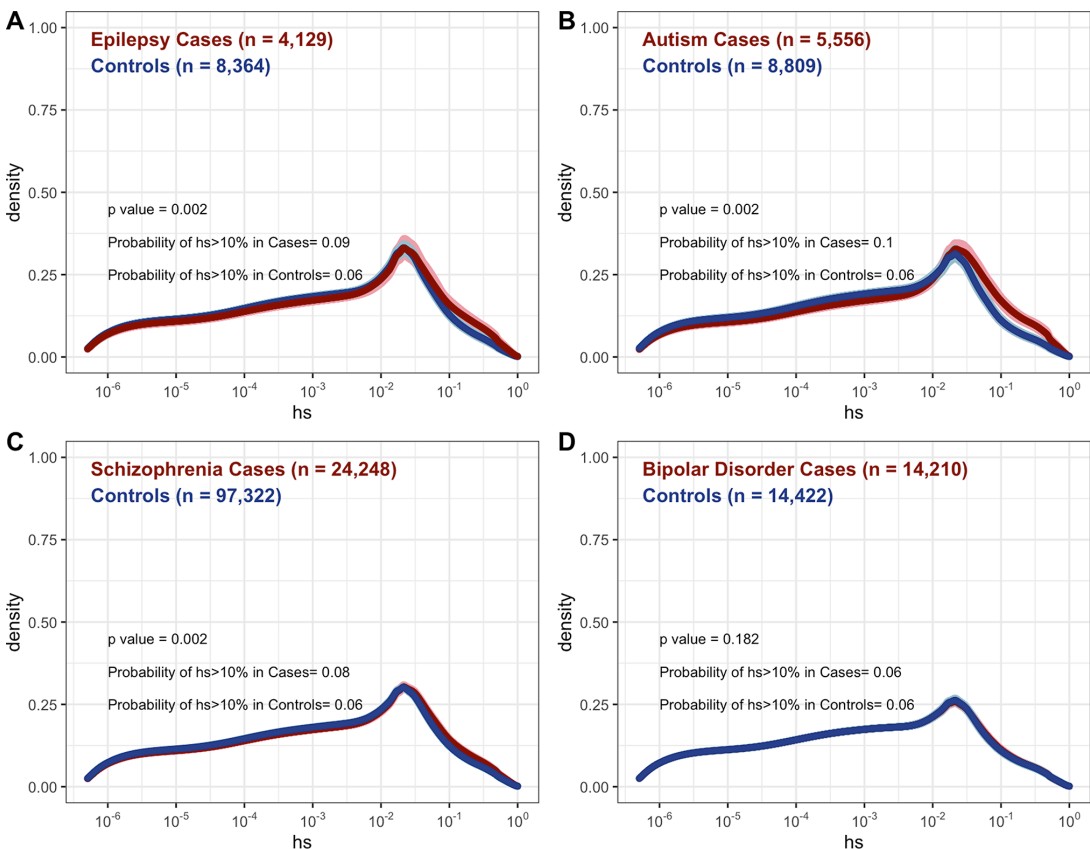

**Appendix 1—figure 11.** Distribution of fitness effects (DFE) for autosomal loss-of-function mutations seen segregating in cases (red curves) versus controls (blue curves) for (**A**) epilepsy (*Feng et al., 2019*), (**B**) autism (*Satterstrom et al., 2020*), (**C**) schizophrenia (*Singh et al., 2022*), and (**D**) bipolar disorder (*Palmer et al., 2022*). Counts were only available for rare variants in each cohort, where rare was defined by the original study.

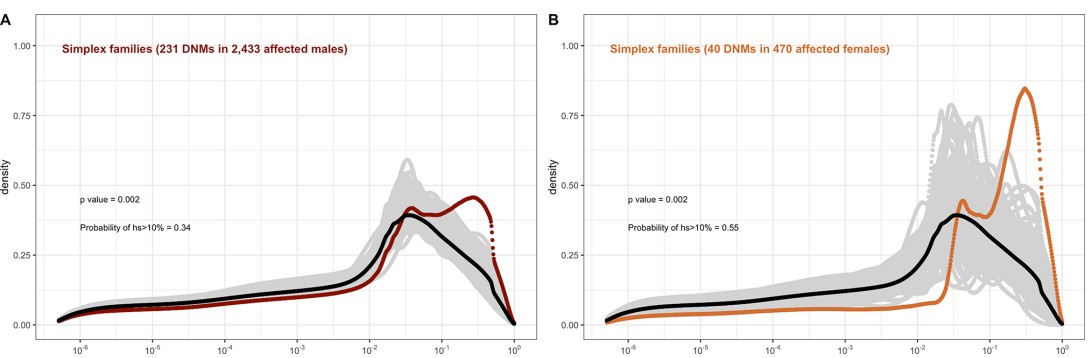

**Appendix 1—figure 12.** The effect of sex of the proband on the fitness effects of de novo mutations (DNMs) seen in simplex families in autism, combining the trios from the Simons Simplex dataset with the simplex families in MSSNG, and with trios in the SPARK study, of which only a small proportion are known to be from multiplex families. In each panel, the distribution of fitness effects (DFE) of all possible loss-of-function (LOF) mutations is denoted with a black curve. For *n* DNMs in a disease cohort, the gray lines denote 100 bootstrapped DFEs of a set of *n* DNMs randomly sampled from the full set of LOF mutational opportunities. The estimated DFE of de novo LOF mutations in (**A**) affected males and (**B**) affected females.

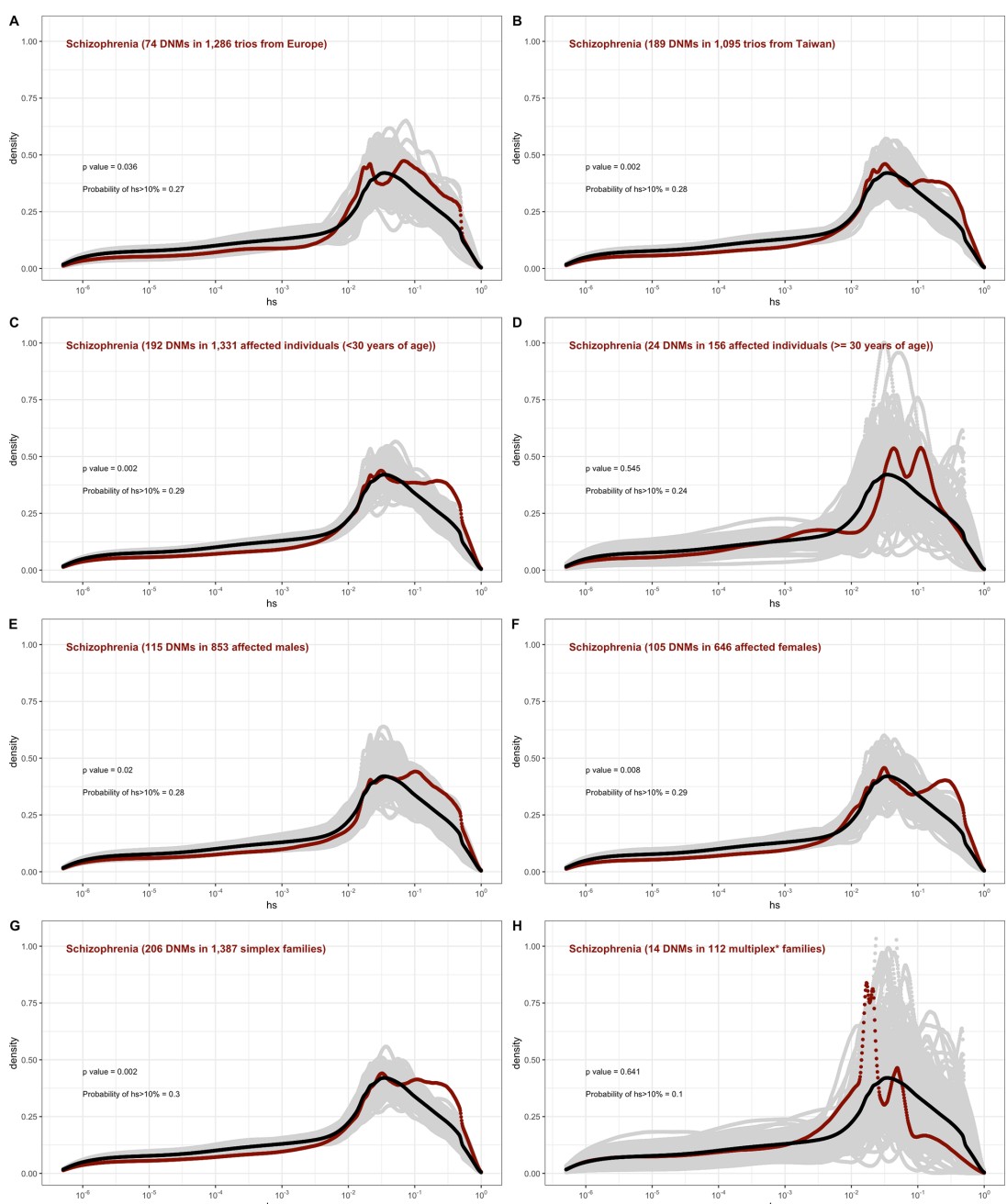

**Appendix 1—figure 13.** The effect of study design and composition on the fitness effects of de novo mutations (DNMs) seen in schizophrenia. In each panel, the distribution of fitness effects (DFE) of all possible loss-of-function (LOF) mutations is denoted with a black curve. For *n* DNMs in a disease cohort, the gray lines denote 100 bootstrapped DFEs of a set of *n* DNMs randomly sampled from the full set of LOF mutational opportunities. The estimated DFE of de novo LOF mutations in (**A**) affected individuals in a European sample, (**B**) affected individuals sampled in Taiwan, (**C**) affected individuals with age of onset reported as less than 30 years of age, (**D**) affected individuals with age of onset reported as at least 30 years of age, (**E**) affected males, (**F**) affected females, (**G**) affected individuals with no reported family history of schizophrenia or other mental illness, and (**I**) affected individuals with a family history of schizophrenia or other mental illness reported.

