## [Editor Report]

This paper directly estimates the fitness cost of loss-of-function mutations in almost every gene in the human genome, providing an interpretable measure of the severity of mutations. The authors then compare datasets of presumably healthy individuals and individuals affected by severe complex disorders or genetic disorders, finding enrichment of de novo loss-of-function mutations in highly constrained genes among probands alongside other illuminating results. This important study will be useful to researchers interested in interpreting and prioritizing disease-causing mutations and in the process of human evolution. Overall, the approach is elegant and the results are of high quality and compelling.

---

## [Decision Letter]

**Decision letter after peer review:**

Thank you for submitting your article "Relating pathogenic loss-of-function mutations in humans to their evolutionary fitness costs" for consideration by *eLife*. Your article has been reviewed by 3 peer reviewers, and the evaluation has been overseen by George Perry as the Senior Editor. The reviewers have opted to remain anonymous.

Essential revisions:

All reviewers were overall strongly positive about your manuscript (and I concur!). The reviewers described your method as "elegant" in their non-public assessments of the work. There are only a few points of revision that you must address in order for your paper to be accepted for publication in *eLife*. These are summarized at a high level immediately below, but please see the full reviews for detailed comments to aid you in your revision. Overall, very well done on this work and paper.

1. Provide some quantification of how robust your analysis is to the choice of parameters including priors, mutation rates, and demographic history.

2. Explicitly expand the discussion of your results in the context of experimental organism-based knowledge of haploinsufficiency and the evolutionary genomics of gene dosage effects.

3. Address the two technical questions from Reviewer #3.

*Reviewer #1 (Recommendations for the authors):*

1) How robust are the results of the demographic model? The authors modify the Schiffels and Durbin model to better match the frequency spectrum but it still doesn't match perfectly. This model also assumes continuous panmictic populations – i.e. it does not take into account admixture or population structure in the history of the sample. While some of these effects may be taken into account by the changes in Ne, I think (but I do not know) that may not capture all the effects. In general, it would be good to have some sense of how the estimates change. I think that in the case of no change in population size, the point estimate of hs is just mu/f, so even just comparing to this would give some idea of how important the demography is.

2) How robust are the results of the choice of prior? The uniform priors on log(s) and h are quite strong statements about your belief about the distribution of effects. The large credible intervals suggest that many genes do not have much posterior information. Would it not make more sense to use priors based on a previous analysis (for example Weghorn et al. (2019))? In any case, I think it is important to know how robust the results are to the priors.

3) How robust are the results of the mutation rate estimates? I'm not totally clear on how these are generated. If I understand correctly, they are trained in part on another dataset of LoF mutations. Is there some circularity there? In particular, if there is some systematic bias in the mutation rate estimates due to variant calling, mapping, or other artefacts would that affect your results? Otherwise, I assume that the random error has a fairly linear effect on your estimates, but it would be good to have a sense of how large that is for different genes – presumably for small genes that are highly constrained, the error in the estimate of mutation rate is systematically larger for example. I don't think it's necessarily needed, but this uncertainty could easily be built into the inference (that is one of the nice things about this approach).

*Reviewer #2 (Recommendations for the authors):*

This study models the fitness costs of loss-of-function mutations in a large cohort of a human database of 55,855 individuals. The modeling indicates different values for autosomal genes, X-linked genes, and those present in the pseudo-autosomal regions of the X and Y chromosomes. The study details the frequency of de novo mutations in zygotes and examined the relationship to a few specific genetic diseases. The authors have composed a well-written manuscript, have explicitly detailed their assumptions, and have noted caveats to interpretations. The results are a valuable documentation of the effects of loss-of-function mutations in humans.

It is perhaps a matter of taste, but this reviewer suggests that the results could have an even greater impact if they were cast in relation to results from experimental organisms regarding haploinsufficiency and to evolutionary genomics of gene dosage effects.

For example, one of the first compilations of LOF mutations that produce a recognizable haploinsufficient effect on a single phenotype revealed that they were typically some type of regulatory gene using *Drosophila* as a model (Birchler et al., 2001, Dev. Biol. 234: 275-288). Experimental studies of LOF heterozygotes in yeast that are haploinsufficient suggest an overrepresentation of genes involved with multicomponent complexes (Papp et al., 2003, Nature 424: 194-197; Deutschbauer et al., 2005, Genetics 169: 1915-1925; Castrillo et al., 2007, J. of Biology 6:4; Yoshikawa et al., 2011, Yeast 28: 349-361; Pir et al., 2012, BMC Systems Biol 6:4). In human, haploinsufficient genes have been associated with those that have strong network connectivity (Huang et al., 2010, PLoS Genetics 6:e1001154; Mottes et al., 2021, PLoS Comput. Biol 17: e1009638) and with connections to human diseases (Makino and McLysaght, 2010, PNAS 107: 9270-9274). Reviews of dosage sensitivity in the context across biology provide examples in other taxa and the role of dosage sensitivity on genomic evolution (Birchler and Veitia, 2012, PNAS 109: 14746-14753; Birchler and Veitia, 2021, Cytogenetics and Genome Research 161:10-11). Is it possible to make these connections?

With regard to the X and Y chromosomes, evidence has been presented that dosage-sensitive regulatory genes were retained between the X and Y chromosomes in the evolution of the heteromorphic state of human sex chromosomes (Bellott et al., 2014, Nature 508: 494-499). It would be of interest to know how the authors' results intersect with that observation. Also, it is of interest how these results fit with the conclusions of Pessia et al. (2012, PNAS 109:5346-5391) regarding how dosage-sensitive genes impacted the evolution of the human sex chromosomes.

The question arises of how the authors' results intersect with findings about dosage sensitivity being preferentially associated with regulatory factors and other multi-component interactions. Do these have a stronger effect? Is there a relationship with the degree of network connectivity? Can the fitness projections help understand their behavior over evolutionary time?

*Reviewer #3 (Recommendations for the authors):*

The method is presented very clearly. I only have two quick technical questions and one (hopefully helpful) suggestion.

1) It seems from the text that h and s have two separate priors and are estimated independently. For rare variants, there is probably power to only estimate their product. Please explain why h and s are treated independently or clarify the text if they are not.

2) Parameter α (male bias) is of significant importance for the analysis of the X chromosome. How is it determined for individual genes (I assume that it is not expected to be 3.5 for every gene)? Could you please clarify?

3) In the deterministic limit and assuming that selection is acting through a disease, the fraction of de novo PTV mutations in patients (out of all PTVs in patients) should be exactly equal selection coefficient. It would be of great interest to compare the population genetics estimates to de novo fractions in the patient population data.

---

## [Author Response]

We greatly appreciate the reviewers’ enthusiasm for the manuscript and thank them for their helpful comments. To address the main comments, we now:

Provide a comparison of results obtained under our demographic model and under a constant population model. We also provide more explicit justification for our choice of prior on *h* and *s* and discuss the effects of mis-specification of the genic mutation rate.After reading the references kindly provided by Reviewer 2, we revised the text to outline how such functional information could be used in extensions of our approach, in order to specify an informative prior for each gene.We now clarify the points about separate priors on *h* and *s*, and our assumptions about alpha in the Materials an methods.

In addition, we have revised the text to address additional suggestions made by reviewers. Detailed point-by-point responses are below.

Essential revisions:Reviewer #1 (Recommendations for the authors):1) How robust are the results of the demographic model? The authors modify the Schiffels and Durbin model to better match the frequency spectrum but it still doesn't match perfectly. This model also assumes continuous panmictic populations – i.e. it does not take into account admixture or population structure in the history of the sample. While some of these effects may be taken into account by the changes in Ne, I think (but I do not know) that may not capture all the effects. In general, it would be good to have some sense of how the estimates change. I think that in the case of no change in population size, the point estimate of hs is just mu/f, so even just comparing to this would give some idea of how important the demography is.

In a constant population size, the expected allele frequency (*f*) for a dominant mutation is *mu*/*hs*; for more weakly selected alleles, the variance can be considerable due to genetic drift (reviewed in Fuller et al. 2019). More recent theory shows that recent changes in population size in human evolution should lead to the same expectation (Simons et al. 2014). These results were verified in simulations by Weghorn et al. (2019), who found that the effects of demography and drift have a larger impact for genes under weak selection, while these effects are negligible for genes under very strong selection.

To illustrate this point, in Author response image 1, we show estimated posterior distributions for 9 genes (of which we estimated strong selection for three, moderate selection for three, and weak selection for three), with circles representing the mode and bars indicating the 95% credible interval. The color of the bar depicts the demographic model under which the posterior distributions were inferred.

**Author response image 1. sa2fig1:** 

As can be seen, the qualitative conclusions about which genes are under strong selection are fairly robust to model misspecification, but the uncertainty can vary considerably depending on the model, and is larger in settings with greater genetic drift.Population structure may also affect allele dynamics: for example, in the presence of strong and sustained inbreeding, recessive mutations will be exposed more often than in a constant population size setting. We have now added a sentence in the main text to emphasize that estimates of selection are sensitive to the choice of the demographic model, especially when the impact of drift is not negligible.

2) How robust are the results of the choice of prior? The uniform priors on log(s) and h are quite strong statements about your belief about the distribution of effects. The large credible intervals suggest that many genes do not have much posterior information. Would it not make more sense to use priors based on a previous analysis (for example Weghorn et al. (2019))? In any case, I think it is important to know how robust the results are to the priors.

We agree with the reviewer that the choice of prior is important, particularly for genes where there is little information (even at this sample size). With this consideration in mind, we deliberately chose a prior that is close to uninformative on the order of magnitude of the strength of selection.

In turn, we do not believe it would make sense to use existing estimates as prior information, e.g. *pLI* (Karczewski et al. 2020)or shet (Weghorn et al., 2019), since they rely on the same signal in overlapping data, and have methodological limitations that we have aimed to overcome here. One way forward may be instead to use an empirical Bayes approach, first pooling information across all genes to obtain a prior, and then inferring posteriors for individual genes. We also envision other extensions of the work in which functional information about genes, independent of the signal of constraint in the data, could be used to specify a prior (some examples of this approach already exist for plants, e.g., Ramstein and Buckler 2022). However, we believe these extensions to be beyond the scope of this paper.

3) How robust are the results of the mutation rate estimates? I'm not totally clear on how these are generated. If I understand correctly, they are trained in part on another dataset of LoF mutations. Is there some circularity there?

We apologize for the confusion: the mutation rates are not derived from the number of LoF mutations. Instead, the gnomAD mutation model on which we rely involves calculating the mutability of each mutation type in its trinucleotide context (e.g., CTG>CAG) based on how often it is seen at intergenic, non-conserved sites in a subset of gnomAD individuals. In each gene, the mutation rate to LOF is then calculated by summing the mutation probabilities of all possible LOF mutations in that gene (Karczewski et al. 2020). We check that the mutation model provides a good fit to de novo mutation data in aggregate (Appendix 1-figure 1). We have now revised the text to hopefully clarify this point.

In particular, if there is some systematic bias in the mutation rate estimates due to variant calling, mapping, or other artefacts would that affect your results? Otherwise, I assume that the random error has a fairly linear effect on your estimates, but it would be good to have a sense of how large that is for different genes – presumably for small genes that are highly constrained, the error in the estimate of mutation rate is systematically larger for example. I don't think it's necessarily needed, but this uncertainty could easily be built into the inference (that is one of the nice things about this approach).

The estimates of the mutation rate should be independent of the constraint on the gene.

However, as noted by the reviewer, errors should have a greater proportional effect on the mutation rate estimates for small genes, as the mutation rate to LOF for a gene is obtained by summing across all LOF mutational opportunities in the gene. For genes under strong selection, as the reviewer notes, mis-specifying the mutation rate by a factor *w* will lead to a *w*-fold bias in the estimated selection coefficient on average.

We do not expect widespread systematic errors in the mutation rate estimates due to variant calling and mapping, since the gnomAD study accounts for various coverage and quality criteria, and the model fits de novo data well in aggregate (Appendix 1-figure 1). While we use only LOF sites annotated as “high confidence” based on functional information (e.g. position in exon), incorrect annotation of some LOF sites may be one source of systematic error, which would affect both mutation rates and observed allele frequencies.

Random error in mutation rate estimates would make the posterior distributions wider, and could be modeled with a prior on mu. We now list this possible extension of our approach in the Discussion.

Reviewer #2 (Recommendations for the authors):This study models the fitness costs of loss-of-function mutations in a large cohort of a human database of 55,855 individuals. The modeling indicates different values for autosomal genes, X-linked genes, and those present in the pseudo-autosomal regions of the X and Y chromosomes. The study details the frequency of de novo mutations in zygotes and examined the relationship to a few specific genetic diseases. The authors have composed a well-written manuscript, have explicitly detailed their assumptions, and have noted caveats to interpretations. The results are a valuable documentation of the effects of loss-of-function mutations in humans.It is perhaps a matter of taste, but this reviewer suggests that the results could have an even greater impact if they were cast in relation to results from experimental organisms regarding haploinsufficiency and to evolutionary genomics of gene dosage effects.For example, one of the first compilations of LOF mutations that produce a recognizable haploinsufficient effect on a single phenotype revealed that they were typically some type of regulatory gene using *Drosophila* as a model (Birchler et al., 2001, Dev. Biol. 234: 275-288). Experimental studies of LOF heterozygotes in yeast that are haploinsufficient suggest an overrepresentation of genes involved with multicomponent complexes (Papp et al., 2003, Nature 424: 194-197; Deutschbauer et al., 2005, Genetics 169: 1915-1925; Castrillo et al., 2007, J. of Biology 6:4; Yoshikawa et al., 2011, Yeast 28: 349-361; Pir et al., 2012, BMC Systems Biol 6:4). In human, haploinsufficient genes have been associated with those that have strong network connectivity (Huang et al., 2010, PLoS Genetics 6:e1001154; Mottes et al., 2021, PLoS Comput. Biol 17: e1009638) and with connections to human diseases (Makino and McLysaght, 2010, PNAS 107: 9270-9274). Reviews of dosage sensitivity in the context across biology provide examples in other taxa and the role of dosage sensitivity on genomic evolution (Birchler and Veitia, 2012, PNAS 109: 14746-14753; Birchler and Veitia, 2021, Cytogenetics and Genome Research 161:10-11). Is it possible to make these connections?With regard to the X and Y chromosomes, evidence has been presented that dosage-sensitive regulatory genes were retained between the X and Y chromosomes in the evolution of the heteromorphic state of human sex chromosomes (Bellott et al., 2014, Nature 508: 494-499). It would be of interest to know how the authors' results intersect with that observation. Also, it is of interest how these results fit with the conclusions of Pessia et al. (2012, PNAS 109:5346-5391) regarding how dosage-sensitive genes impacted the evolution of the human sex chromosomes.The question arises of how the authors' results intersect with findings about dosage sensitivity being preferentially associated with regulatory factors and other multi-component interactions. Do these have a stronger effect? Is there a relationship with the degree of network connectivity? Can the fitness projections help understand their behavior over evolutionary time?

We thank the reviewer for raising several interesting points, and for the associated references.

Genes that are haploinsufficient with respect to phenotypes highly correlated with fitness, such as early-onset severe diseases, are expected to also be enriched among genes under strong selection. There are many such examples among Mendelian disease genes, and the link is explicitly assumed in the naming of *pLI* (“probability of being LoF intolerant”) (e.g., Cassa et al. 2017; Lek et al. 2016).

In turn, several studies provided by the reviewer support the notion that the dosage-sensitivity of genes, in particular components of multi-protein complexes that have stoichiometric constraints on the number of copies, has influenced the evolution of gene duplicates and patterns of transcriptional silencing. On the X chromosome, it is suggested that important regulatory genes that require similar dosages in both sexes and matched doses between the X and autosomes (e.g., because they are part of protein complexes with autosomal components) have approximately doubled their expression levels per copy, or in many cases retained a functional homolog on the Y chromosome (Pessia et al. 2012; Bellott et al. 2014).

While it must be the case that genes with important functions sensitive to dosage have evolved various dosage compensation mechanisms, it is not always clear how the observed dosage of a gene at present relates to the fitness consequence of losing a copy. Genes with expression from two copies in both sexes (San Roman et al. 2022) outside the pseudo-autosomal region (PAR) tend to have very high *hs* estimates, as expected if having balanced X:A dosage in both sexes is characteristic of functionally critical genes; moreover, many of these genes are part of large multi-subunit complexes (Pessia et al. 2012; Bellott et al. 2014). Less readily explained is the observation that many X chromosome genes with one copy expressed in males and females apiece and imbalanced X:A dosage (San Roman et al. 2022) have comparably high *hs* estimates (Figure S8A). Also unexpected is that genes in the PAR, which have two expressed copies in males and females, on average have much smaller *hs* estimates than other genes on the X and autosomes (Figure S8A).

Following the reviewer’s suggestion to investigate the link between *hs* and the network connectivity of genes, we built a network graph of protein interactions in the human BioGRID database (Oughtred et al. 2021) and counted the number of edges or “interactions” for each protein. We find that, as they expected, *hs* is positively correlated with the number of connections in the protein-interaction network (R^2^ = 4%; p < 2e-16; see also Kim et al. 2019). However, this finding alone is difficult to interpret: in particular, it may be due at least in part to ascertainment in current network datasets, where genes that are of interest for a disease are more extensively studied, so tend to have more known interactions.

While learning about particular functional properties or evolutionary events through their relationship to fitness effects is challenging, there are nonetheless ways to synthesize functional and fitness measures. One natural extension of our model would be using expression levels, dosage sensitivity, and gene network information to specify gene-specific priors, or priors for categories of genes (some examples of this approach already exist for plants, e.g., Ramstein and Buckler 2022). While beyond the scope of this study, such approaches may become the norm as tissue-specific and cell-specific functional information becomes more widely available. Following the reviewer’s comments, we now make some of these points more explicitly in our discussion.

Reviewer #3 (Recommendations for the authors):The method is presented very clearly. I only have two quick technical questions and one (hopefully helpful) suggestion.

We thank the reviewer for their generous feedback.

1) It seems from the text that h and s have two separate priors and are estimated independently. For rare variants, there is probably power to only estimate their product. Please explain why h and s are treated independently or clarify the text if they are not.

We apologize for being unclear, and have added a clarifying sentence to the Methods.

When there is selection on heterozygotes (i.e., other than for completely recessive alleles), and assuming random-mating, alleles do not reach high enough frequency to be found in homozygotes and selection is in heterozygotes. Consequently, for semi-dominant and dominant mutations, we can only learn about the compound parameter *hs* (see, e.g., Fuller et al. 2019). We nonetheless use separate priors for *h* and *s* so that we have the same priors for genes on the X (where we propose *h* separately for females) and the autosomes and can more reliably compare the estimates for X and autosomes.

2) Parameter α (male bias) is of significant importance for the analysis of the X chromosome. How is it determined for individual genes (I assume that it is not expected to be 3.5 for every gene)? Could you please clarify?

We are assuming that the mutation rate on the X is given by α, and the same mutation model for all genes on the X. While an approximation, we believe this choice is justified because (1) the sex bias in mutation rates varies little among mutation types (see Gao et al. 2019; Jónsson et al. 2017), ranging from 3-4 for parents of similar ages. (2) In principle, there could be additional factors, such as replication timing or expression levels, which lead to different sex biases in mutation for different genes. In practice, this does not appear to be a big effect. For example, a good predictor of the total mutation rate of a gene is simply its length. More generally, there is relatively little variation in mutation rates beyond the scale of 10 bps (Aggarwala and Voight 2016; Seplyarskiy and Sunyaev 2021). We now add a sentence noting this consideration in the Methods.

3) In the deterministic limit and assuming that selection is acting through a disease, the fraction of de novo PTV mutations in patients (out of all PTVs in patients) should be exactly equal selection coefficient. It would be of great interest to compare the population genetics estimates to de novo fractions in the patient population data.

We agree that such a comparison is of great interest, and it was in fact included in a nice analysis by Weghorn et al., 2019 (their Figure 2). Unfortunately, we are limited by the kinds of data that are publicly available: for most individuals for whom de novo mutations are publicly available, the full set of mutations they carry is not.

References

Aggarwala, Varun, and Benjamin F. Voight. 2016. “An Expanded Sequence Context Model Broadly Explains Variability in Polymorphism Levels across the Human Genome.” Nature Genetics 48 (4): 349–55.

Bellott, Daniel W., Jennifer F. Hughes, Helen Skaletsky, Laura G. Brown, Tatyana Pyntikova, Ting-Jan Cho, Natalia Koutseva, et al. 2014. “Mammalian Y Chromosomes Retain Widely Expressed Dosage-Sensitive Regulators.” Nature 508 (7497): 494–99.

Cassa, Christopher A., Donate Weghorn, Daniel J. Balick, Daniel M. Jordan, David Nusinow, Kaitlin E. Samocha, Anne O’Donnell-Luria, et al. 2017. “Estimating the Selective Effects of Heterozygous Protein-Truncating Variants from Human Exome Data.” Nature Genetics 49 (5): 806–10.

Fuller, Zachary L., Jeremy J. Berg, Hakhamanesh Mostafavi, Guy Sella, and Molly Przeworski. 2019. “Measuring Intolerance to Mutation in Human Genetics.” Nature Genetics 51 (5): 772–76.

Gao, Ziyue, Priya Moorjani, Thomas A. Sasani, Brent S. Pedersen, Aaron R. Quinlan, Lynn B. Jorde, Guy Amster, and Molly Przeworski. 2019. “Overlooked Roles of DNA Damage and Maternal Age in Generating Human Germline Mutations.” Proceedings of the National Academy of Sciences of the United States of America 116 (19): 9491–9500.

Jónsson, Hákon, Patrick Sulem, Birte Kehr, Snaedis Kristmundsdottir, Florian Zink, Eirikur Hjartarson, Marteinn T. Hardarson, et al. 2017. “Parental Influence on Human Germline de Novo Mutations in 1,548 Trios from Iceland.” Nature 549 (7673): 519–22.

Karczewski, Konrad J., Laurent C. Francioli, Grace Tiao, Beryl B. Cummings, Jessica Alföldi, Qingbo Wang, Ryan L. Collins, et al. 2020. “The Mutational Constraint Spectrum Quantified from Variation in 141,456 Humans.” Nature 581 (7809): 434–43.

Kim, Samuel S., Chengzhen Dai, Farhad Hormozdiari, Bryce van de Geijn, Steven Gazal, Yongjin Park, Luke O’Connor, et al. 2019. “Genes with High Network Connectivity Are Enriched for Disease Heritability.” American Journal of Human Genetics 104 (5): 896–913.

Lek, Monkol, Konrad J. Karczewski, Eric V. Minikel, Kaitlin E. Samocha, Eric Banks, Timothy Fennell, Anne H. O’Donnell-Luria, et al. 2016. “Analysis of Protein-Coding Genetic Variation in 60,706 Humans.” Nature 536 (7616): 285–91.

Oughtred, Rose, Jennifer Rust, Christie Chang, Bobby-Joe Breitkreutz, Chris Stark, Andrew Willems, Lorrie Boucher, et al. 2021. “The BioGRID Database: A Comprehensive Biomedical Resource of Curated Protein, Genetic, and Chemical Interactions.” Protein Science: A Publication of the Protein Society 30 (1): 187–200.

Pessia, Eugénie, Takashi Makino, Marc Bailly-Bechet, Aoife McLysaght, and Gabriel A. B. Marais. 2012. “Mammalian X Chromosome Inactivation Evolved as a Dosage-Compensation Mechanism for Dosage-Sensitive Genes on the X Chromosome.” Proceedings of the National Academy of Sciences of the United States of America 109 (14): 5346–51.

Ramstein, Guillaume P., and Edward S. Buckler. 2022. “Prediction of Evolutionary Constraint by Genomic Annotations Improves Functional Prioritization of Genomic Variants in Maize.” Genome Biology 23 (1): 183.

San Roman, Adrianna K., Alexander K. Godfrey, Helen Skaletsky, Daniel W. Bellott, Abigail F. Groff, Hannah L. Harris, Laura V. Blanton, et al. 2022. “The Human Inactive X Chromosome Modulates Expression of the Active X Chromosome.” bioRxiv. https://doi.org/10.1101/2021.08.09.455676.

Seplyarskiy, Vladimir B., and Shamil Sunyaev. 2021. “The Origin of Human Mutation in Light of Genomic Data.” Nature Reviews. Genetics, June, 1–15.

Simons, Yuval B., Michael C. Turchin, Jonathan K. Pritchard, and Guy Sella. 2014. “The Deleterious Mutation Load Is Insensitive to Recent Population History.” Nature Genetics 46 (3): 220–24.

Turner, Tychele N., Amy B. Wilfert, Trygve E. Bakken, Raphael A. Bernier, Micah R. Pepper, Zhancheng Zhang, Rebecca I. Torene, Kyle Retterer, and Evan E. Eichler. 2019. “Sex-Based Analysis of De Novo Variants in Neurodevelopmental Disorders.” American Journal of Human Genetics 105 (6): 1274–85.

Weghorn, Donate, Daniel J. Balick, Christopher Cassa, Jack A. Kosmicki, Mark J. Daly, David R. Beier, and Shamil R. Sunyaev. 2019. “Applicability of the Mutation–Selection Balance Model to Population Genetics of Heterozygous Protein-Truncating Variants in Humans.” Molecular Biology and Evolution 36 (8): 1701–10.